# Integrative genomic and transcriptomic analysis of leiomyosarcoma

Priya Chudasama

Leiomyosarcoma (LMS) is an aggressive mesenchymal malignancy with few therapeutic options. The mechanisms underlying LMS development, including clinically actionable genetic vulnerabilities, are largely unknown. Here we show, using whole-exome and transcriptome sequencing, that LMS tumors are characterized by substantial mutational heterogeneity, near-universal inactivation of *TP53* and *RB1*, widespread DNA copy number alterations including chromothripsis, and frequent whole-genome duplication. Furthermore, we detect alternative telomere lengthening in 78% of cases and identify recurrent alterations in telomere maintenance genes such as *ATRX*, *RBL2*, and *SP100*, providing insight into the genetic basis of this mechanism. Finally, most tumors display hallmarks of "BRCAness", including alterations in homologous recombination DNA repair genes, multiple structural rearrangements, and enrichment of specific mutational signatures, and cultured LMS cells are sensitive towards olaparib and cisplatin. This comprehensive study of LMS genomics has uncovered key biological features that may inform future experimental research and enable the design of novel therapies.

---

#A full list of authors and their affliations appears at the end of the paper.

Leiomyosarcomas (LMS) are malignant tumors of smooth-muscle origin that occur across age groups, accounting for 10% of all soft-tissue sarcomas, and most commonly involve the uterus, retroperitoneum, and large blood vessels. Long-term survival in LMS patients may be achieved through surgical excision and adjuvant radiotherapy. However, local recurrence and/or metastasis develop in approximately 40% of cases[1]. Patients with disseminated LMS are usually incurable, as reflected by a median survival after development of distant metastases of 12 months[2], and cytotoxic chemotherapy is generally administered with palliative intent.

Cytogenetic studies have shown that LMS are genetically complex, often exhibiting chaotic karyotypes, and no pathognomonic chromosomal rearrangements have been detected. More recent investigations employing microarray technologies and targeted sequencing approaches have provided insight into recurrent genetic features of LMS and associated histopathologic characteristics and clinical outcomes[3–5]. However, systematic genome- and transcriptome-wide investigations of LMS using next-generation sequencing technology are lacking, and clinically actionable genetic vulnerabilities remain unknown.

In this study, we have used whole-exome and RNA sequencing to characterize the molecular landscape of LMS. We identify a perturbed tumor suppressor network, widespread genomic instability, and alternative lengthening of telomeres (ALT) as hallmarks of this disease. Furthermore, our findings uncover genomic imprints of defective homologous recombination repair (HRR) of DNA double-strand breaks as potential liability of LMS tumors that could be exploited for therapeutic benefit, and provide a map for future studies of additional genetic alterations or deregulated cellular processes as entry points for molecularly targeted interventions.

## Results

**Mutational landscape of LMS.** We performed whole-exome sequencing and transcriptome sequencing in a cohort of 49 patients with LMS (non-uterine, $n = 39$; uterine, $n = 10$; newly diagnosed, $n = 20$; locally recurrent, $n = 6$; metastatic, $n = 23$) (Supplementary Data 1). We detected a total of 14,259 (median, 223; range, 79–1101) somatic single-nucleotide variants (SNVs), of which 2522 (median, 39; range, 10–226) were non-silent, and 297 somatic small insertions/deletions (indels; median, 3; range, 0–50) (Fig. 1a and Supplementary Data 1). The median somatic mutation rate was 3.09 (range, 1.05–14.76) per megabase (Mb) of target sequence, comparable to the rates observed in clear-cell kidney cancer or hepatocellular carcinoma[6]. Recurrence analysis using MutSigCV[7] identified *TP53* (49%), *RB1* (27%), and *ATRX* (24%) as significantly mutated genes ($q < 0.01$, Benjamini–Hochberg correction) (Fig. 1a and Supplementary Figure 1a). *TP53* mutations clustered in the DNA binding and tetramerization motifs, whereas those affecting *RB1* and *ATRX* were distributed across the entire protein (Fig. 1b). SNVs and indels were also present in other established cancer genes[8], albeit at low frequencies (Fig. 1a, Supplementary Figure 1a, and Supplementary Data 1). Network analysis of the integrated collection of SNVs and indels using HotNet2[9] identified two significantly mutated subnetworks centered on *TP53* and *RB1* as "hot" nodes ($P < 0.05$, two-stage multiple hypothesis test and 100 permutations of the global interaction network), which encompassed genes related to DNA damage response and telomere maintenance (*TOPORS*, *ATR*, *TP53BP1*, *TELO2*), cell cycle and apoptosis regulation (*PSDM11*, *CASP7*, *XPO1*), epigenetic regulation (*HIST3H3*, *SETD7*, *KMT2C*), MAPK signaling and positive regulation of muscle cell proliferation (*MAPK14*, *DUSP10*, *MEF2C*), regulation of mRNA stability (*ZFP36L1*, *SRSF5*), and

PI3K-AKT signaling (*MTOR*, *LAMA4*) (Fig. 1c). These data showed that LMS tumors exhibit substantial mutational heterogeneity and are possibly driven by loss of TP53 and/or RB1 function together with a diverse spectrum of less commonly mutated "gene hills", which may be different for each patient[10].

**Widespread DNA copy number changes and chromothripsis in LMS.** We next performed genome-wide analysis of somatic copy-number alterations (CNAs) and identified recurrent losses in regions of chromosomes 10, 11q, 13, 16q, and 17p13 (comprising *TP53*) and recurrent gains of chromosome 17p12 (affecting *MYOCD*) (Fig. 2a, b and Supplementary Figure 1b), consistent with previous molecular cytogenetic studies[5, 11]. Most recurrently mutated genes were additionally targeted by CNAs (Supplementary Figure 1a). Furthermore, multiple cancer drivers as well as components of the CINSARC prognostic gene expression signature[12] were affected by CNAs in at least 30% of cases, including genes encoding tumor suppressors (*PTEN*, *RB1*, *TP53*), DNA repair proteins (*BRCA2*, *ATM*), chromatin modifiers (*RBL2*, *DNMT3A*, *KAT6B*), cytokine receptors (*ALK*, *FGFR2*, *FLT3*, *LIFR*), and transcriptional regulators (*PAX3*, *FOXO1*, *CDX2*, *SUFU*) (Fig. 2a). We also detected regions of significant focal gains and losses using GISTIC2.0[13] ($q < 0.25$, Benjamini–Hochberg correction) (Fig. 2b and Supplementary Data 1), and clustering of broad and focal CNAs demonstrated that individual tumors had highly rearranged genomes (Supplementary Figure 1b). In addition, chromothripsis[14] was present in 17 of 49 samples (35%), with the number of affected chromosomes per tumor ranging from one to five (Fig. 2c and Supplementary Data 1). Thus, variable patterns of widespread CNA and localized chromosome shattering further add to the genomic complexity of LMS.

**Transcriptomic characterization of LMS.** We next sought to delineate biologically relevant subgroups of LMS defined by different gene expression profiles. Both unsupervised hierarchical clustering (Fig. 3a) and principal component analysis (Supplementary Figure 2a) revealed three distinct subgroups of patients. Gene ontology analysis using DAVID on the top 100 highly variable genes showed greater than tenfold enrichment (false discovery rate < 0.05) of biological processes related to platelet degranulation, complement activation, and metabolism for subgroup 1; and muscle development and function and regulation of membrane potential for subgroup 2. Subgroup 3 was characterized by low expression of genes separating subgroups 1 and 2, but showed medium to high levels of genes associated with myofibril assembly, muscle filament function, and cell–cell signaling common to subgroups 1 and 2 (Fig. 3a and Supplementary Data 1). Increased expression of *ARL4C* or *CASQ2* and *LMOD1*, respectively, indicated that subgroups 2 and 3 correspond to previously identified LMS subtypes II and I (Supplementary Figure 2b)[15].

**Biallelic inactivation of *TP53* and *RB1* in LMS.** Further analysis of transcriptome data, coupled with RT-PCR validation, uncovered high-confidence fusion transcripts arising from chromosomal rearrangements in 34 of 37 cases (total number of fusions, $n = 183$; range, $n = 1–29$; Fig. 3b, c, Supplementary Figure 2c, and Supplementary Data 1). While no recurrent fusions were detected, multiple rearrangements targeted *TP53* and *RB1* (Fig. 4a, b), which were predicted to result in out-of-frame fusion proteins or loss of critical functional domains in the majority of cases. This indicated that *TP53* and *RB1* are disrupted by a variety of genetic mechanisms in LMS tumors. In accordance, careful examination of exome data additionally revealed protein-damaging microdeletions (20–100 base pairs (bp)), inversions, and exon skipping

events (Fig. 4c and Supplementary Figure 3a–c). Furthermore, we identified three cases with pathogenic germline alterations affecting *TP53* (hemizygous loss, *n* = 1) or *RB1* (mutation, *n* = 2). Integration of SNVs, indels, CNAs, fusions, and microalterations

demonstrated biallelic disruption of *TP53* and *RB1* in 92 and 94% of cases, respectively (Fig. 5a). Three tumors with wild-type *RB1* displayed loss of *CDKN2A* expression and overexpression of *CCND1* as alternative mechanisms of RB1 suppression (Fig. 5b).

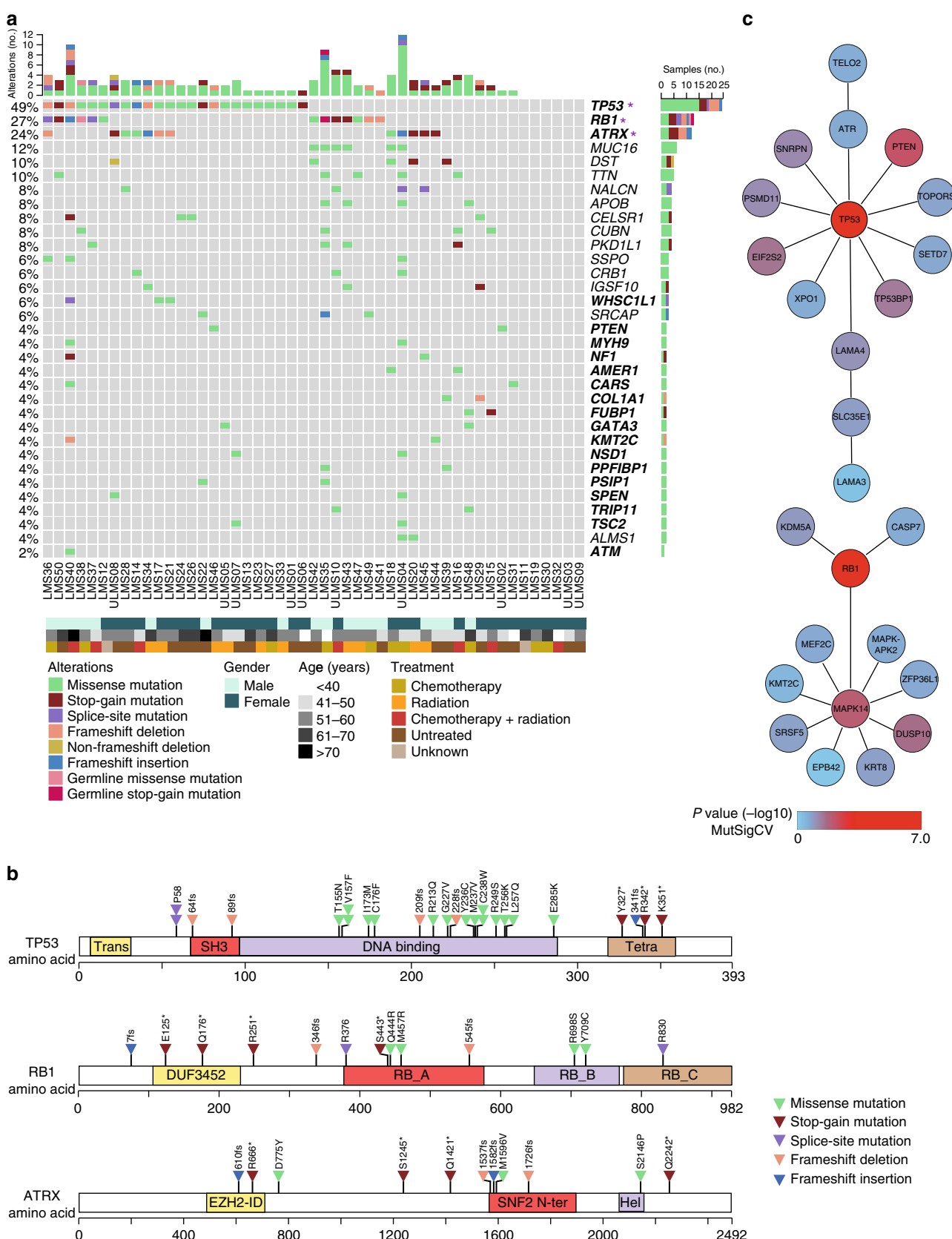

Finally, we detected a single loss-of-function mutation in the basic helix-loop-helix domain of MAX, previously described in hereditary pheochromocytoma[16], that was associated with over-expression of *CDK4* and *CCND2* (Fig. 5b), possibly through enhanced formation of MYC-MAX heterodimers activating the *CDK4* and *CCND2* promoters or via disruption of the MAD-MAX repressor complex[17, 18]. These data showed that inactivation of TP53 and RB1 is near-obligatory in LMS.

**Whole-genome duplication in LMS.** The variant allele frequencies of SNVs and indels affecting *TP53* and *RB1* were congruent with tumor purity, establishing TP53 and RB1 inactivation as truncal events in LMS development (Fig. 5c). Further investigation of allele-specific copy number profiles revealed that 27 of 49 cases had undergone whole-genome duplication (WGD), resulting in an average ploidy close to 4 (Fig. 5a, d and Supplementary Data 1). In most cases, only mutant *TP53* and *RB1* were detectable irrespective of ploidy, suggesting that the respective wild-type alleles had been lost before WGD. Accordingly, the allele-specific copy number profiles for a primary tumor/metastasis pair demonstrated that the former had acquired *TP53* and *RB1* alterations with concomitant loss of wild-type chromosomes 17p and 13 (Fig. 5d). By comparison, the metastasis showed only minor differences regarding mutations and fusion transcripts, but had undergone WGD (Fig. 5d, Supplementary Figure 2c, and Supplementary Data 1), implying that tetraploidization was a progression event preceded by loss of wild-type *TP53* and *RB1*. These data again indicated that LMS is driven by a perturbed tumor suppressor network (Fig. 5e), which gives rise to WGD and gross genomic instability, thereby accelerating tumor evolution, in the majority of cases[19–21].

**High frequency of ALT in LMS.** To achieve replicative immortality, approximately 85% of cancers re-activate *TERT* expression[22]. The remaining 15% maintain telomere length via a telomerase-independent mechanism termed ALT, which appears to be particularly prevalent in cells of mesenchymal origin[23–25]. ALT has been correlated with loss of ATRX, a chromatin remodeling factor that incorporates histone variant H3.3 into telomeric and pericentromeric regions in complex with DAXX[26–28]. Our finding of recurrent *ATRX* alterations (SNVs, indels, CNAs; Fig. 1a, b and Supplementary Figure 1a) suggested that ALT might be a common feature of LMS. We therefore tested 49 patient samples for the presence of C-circles, extrachromosomal telomeric repeats that are hallmarks of ALT[29]. C-circles were detected in 38 of 49 samples (78%; Fig. 6a and Supplementary Data 1), the highest frequency of ALT reported to date for any tumor entity[30]. ALT-positive cells also display extensive telomere length heterogeneity, including the presence of very long telomeres[31] and typically resulting in high telomere content. Quantitative PCR revealed a wide range of telomere content in both ALT-positive and ALT-negative LMS tumors, but no correlation between ALT status and telomere content, both absolute and relative to normal controls, indicating that telomere content is not a relevant marker for ALT in LMS (Fig. 6b). Since the frequency of ALT considerably exceeded that of potentially deleterious *ATRX* alterations (Figs. 1a, c and 6a, c), we investigated additional genes from the TelNet database and observed that LMS tumors are characterized by recurrent alterations in a broad spectrum of telomere maintenance genes (Fig. 6c). Of these, deletions of *RBL2* ($P = 0.008$) and *SP100* ($P = 0.02$) showed the strongest association with ALT positivity ($P$-values determined by Fisher exact test). RBL2 has been shown to block ALT by interacting with RINT1[32]. SP100 has been implicated in ALT suppression by sequestering the MRE11/RAD50/NBS complex and is a major component of ALT-associated PML bodies[33, 34]. These data indicated that mechanisms beyond ATRX loss account for the exceptionally high frequency of ALT in LMS.

**"BRCAness" as potentially actionable feature of LMS.** Our finding of frequent deletions targeting genes implicated in HRR of DNA double-strand breaks (Fig. 2a), e.g. *ATM*, *BRCA2*, and *PTEN*[35–37], prompted us to inquire if LMS tumors show genomic imprints of defective HRR, i.e. a "BRCAness" phenotype[6, 38–40], which confers sensitivity to DNA double-strand break-inducing drugs, such as platinum derivatives, and poly(ADP-ribose) polymerase (PARP) inhibitors[41]. We first interrogated genes that have been described as synthetic lethal to PARP inhibition[38, 42] and observed deleterious aberrations in multiple HRR components, including *PTEN* (57%), *BRCA2* (53%), *ATM* (22%), *CHEK1* (22%), *XRCC3* (18%), *CHEK2* (12%), *BRCA1* (10%), and *RAD51* (10%), as well as in members of the Fanconi anemia complementation groups, namely *FANCA* (27%) and *FANCD2* (10%) (Fig. 7a). Next, we detected enrichment of five known mutational signatures[6] (Alexandrov-COSMIC (AC) 1: clock-like, spontaneous deamination; AC3: associated with defective HRR; AC5: clock-like, mechanism unknown; AC6 and AC26: associated with mismatch repair (MMR) defects). Signature AC3 contributed to the mutational catalog in 98% of samples, and the confidence interval of the exposure to AC3 excluded zero in 57% of samples (Fig. 7b). Comparison of the signatures identified in the LMS cohort against a background of 7042 cancer samples (whole-genome sequencing, $n = 507$; whole-exome sequencing, $n = 6535$)[6] demonstrated significant enrichment of AC1 ($P = 1.31 \times 10^{-3}$), AC3 ($P = 2.67 \times 10^{-30}$), and AC26 ($P = 9.28 \times 10^{-41}$) in LMS tumors ($P$-values determined by Fisher exact test followed by Benjamini–Hochberg correction). Finally, clonogenic assays demonstrated that LMS cell lines harboring aberrations of multiple genes that are synthetic lethal to PARP inhibition (Supplementary Figure 4) responded to the PARP inhibitor olaparib in a dose-dependent manner, an effect that was enhanced by a pulse of cisplatin prior to continuous olaparib treatment (Fig. 7c). These data showed that most LMS tumors exhibit phenotypic traits of "BRCAness", which might provide a rationale for therapies that target defective HRR.

## Discussion

This study represents a comprehensive analysis of the genomic alterations that underlie the development of LMS, an aggressive and difficult-to-treat malignancy for which no targeted therapy

**Fig. 1** Mutational landscape of adult LMS. **a** Frequency and type of mutations. Rows represent individual genes, columns represent individual tumors. Genes are sorted according to frequency of SNVs/indels (left). Asterisks indicate significantly mutated genes according to MutSigCV. Bars depict the number of SNVs/indels for individual tumors (top) and genes (right). Established cancer genes are shown in bold. Types of mutations and selected clinical features are annotated according to the color codes (bottom). **b** Schematic representation of SNVs/indels in *TP53*, *RB1*, and *ATRX*. Protein domains are indicated (Trans transactivation domain, SH3 Src homology 3-like domain, Tetra tetramerization domain, DUF3452 domain of unknown function, RB_A RB1-associated protein domain A, RB_B RB1-associated protein domain B, RB_C RB1-associated protein domain C, EZH2-ID EZH2 interaction domain, SNF2 N ter SNF2 family N-terminal domain, Hel helicase domain). **c** Top subnetworks from HotNet2 analysis of genes harboring SNVs/indels. MutSigCV $P$-values (−log10) for individual genes are annotated according to the color code

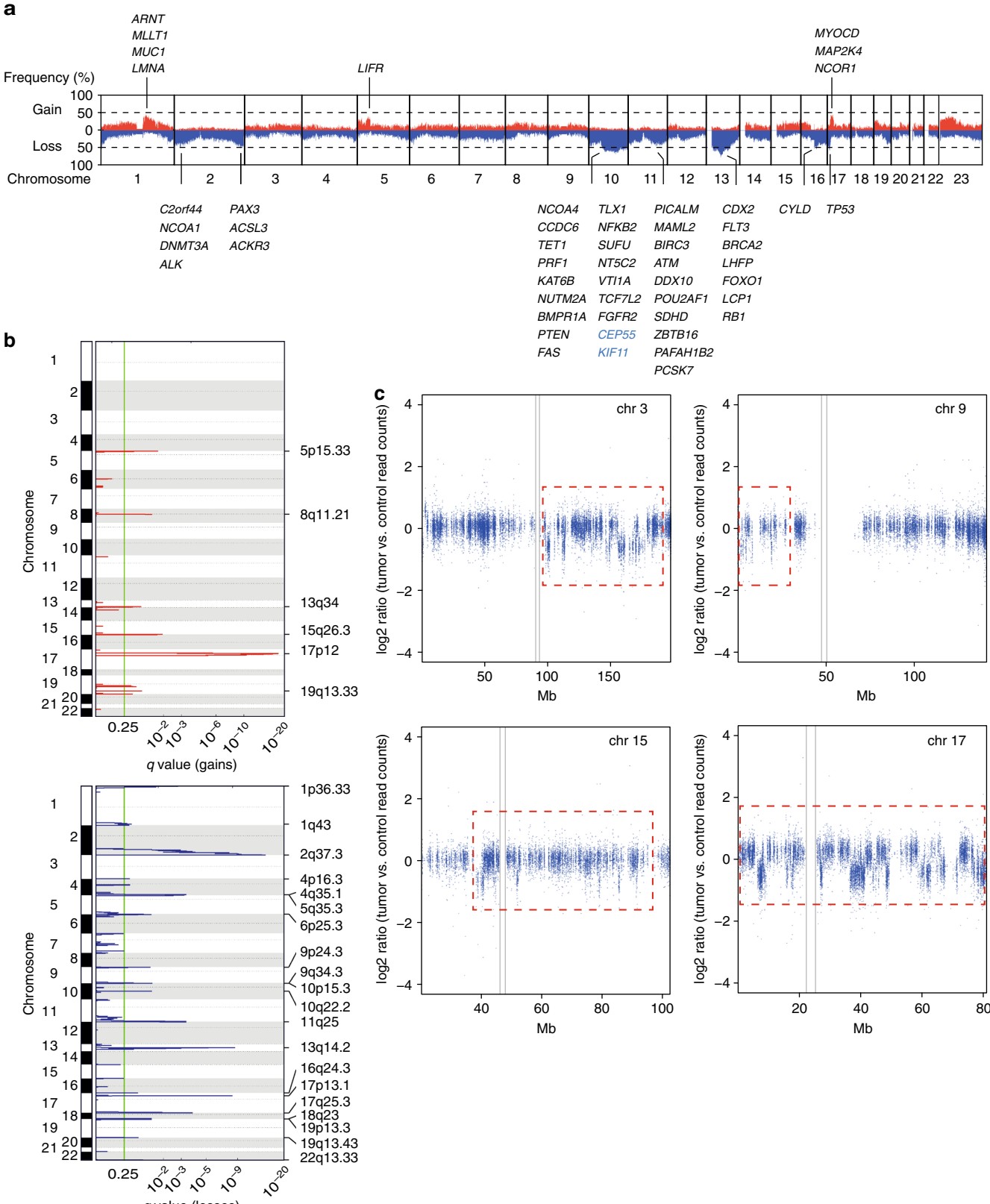

**Fig. 2** Genomic imbalances in adult LMS. **a** Overall pattern of CNAs. Chromosomes are represented along the horizontal axis, frequencies of chromosomal gains (red) and losses (blue) are represented along the vertical axis. Established cancer genes (black) and components of the CINSARC signature (blue) affected by CNAs in at least 30% of cases are indicated. **b** GISTIC2.0 plot of recurrent focal gains (top) and losses (bottom). The green line indicates the cut-off for significance ($q = 0.25$). **c** Read-depth plots of case LMS24 showing oscillating CNAs of chromosomes 3, 9, 15, and 17 (red dotted lines), indicative of chromothripsis. Gray lines indicate centromeres. Mb megabase, chr chromosome

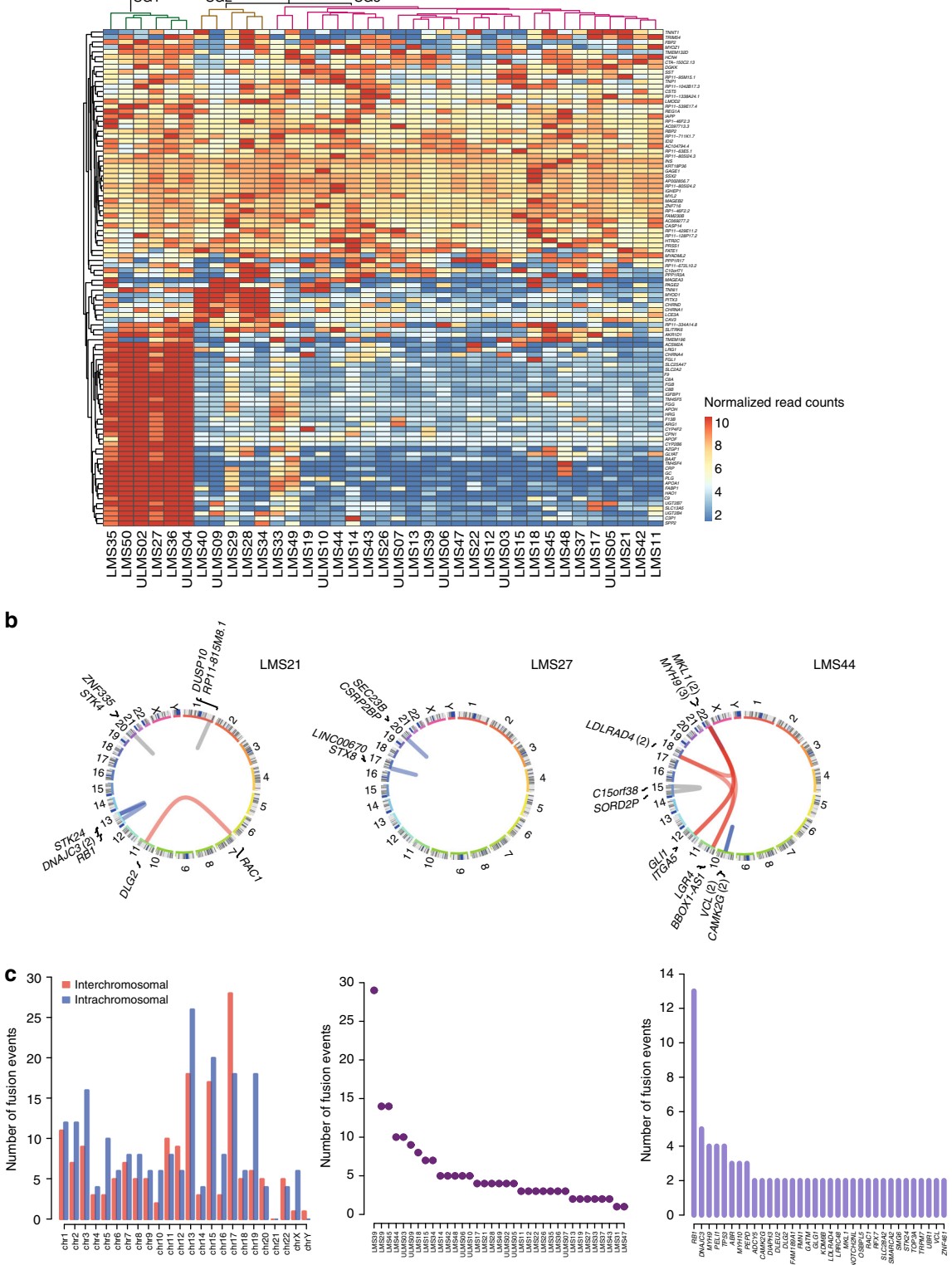

**Fig. 3** Transcriptomic characterization of adult LMS. **a** Unsupervised hierarchical clustering based on the top 100 differentially expressed genes showing separation of tumors into three subgroups (SG1–3; dendrogram colors green, brown, and magenta). The heatmap displays normalized read count values for individual genes, which were centered, scaled (z-score), and quantile-discretized. **b** Structural variant plots of fusion transcripts in three tumors identified by TopHat2 and validated by RT-PCR (blue, intrachromosomal; red, interchromosomal) or visual inspection using Integrative Genomics Viewer (gray). Numbers in parentheses indicate the number of fusions involving the respective gene. **c** Number of fusion events per chromosome (left), tumor (middle), and gene (right). chr, chromosome

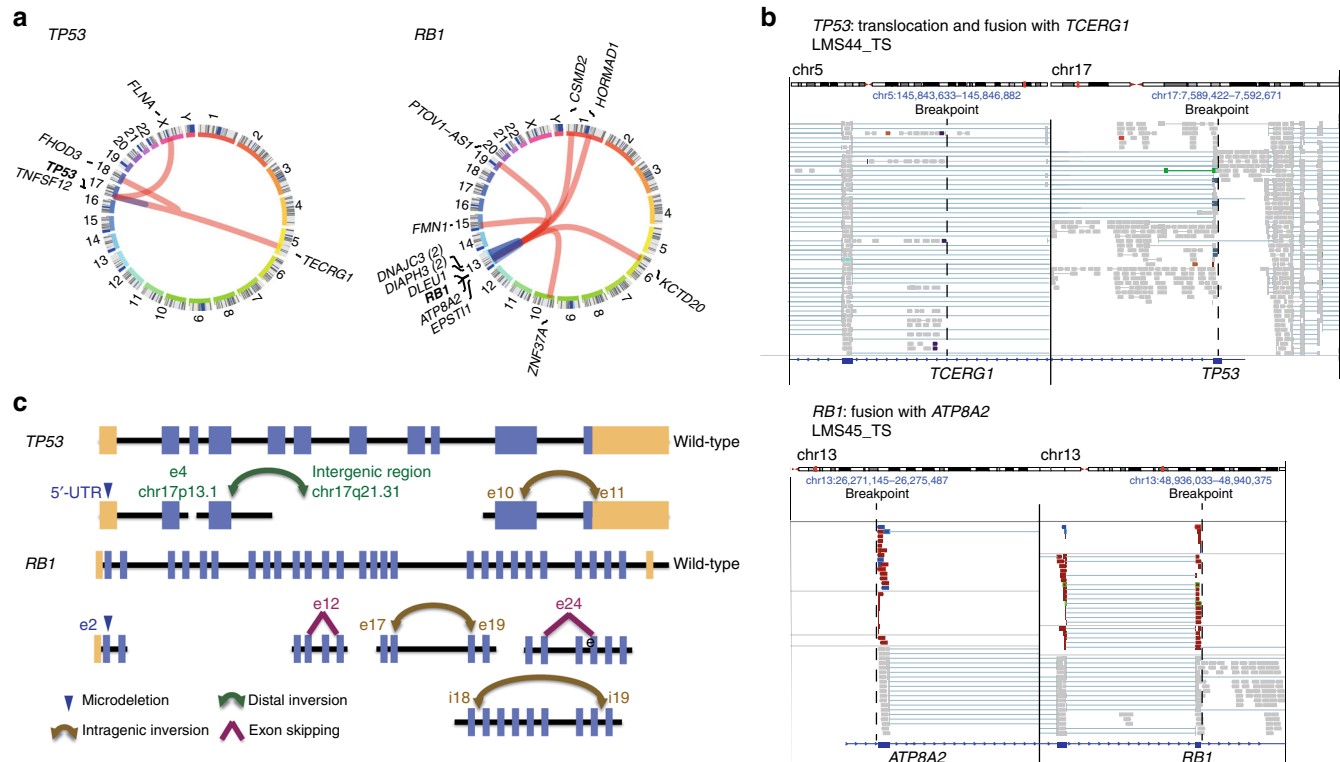

**Fig. 4** Genetic lesions targeting *TP53* and *RB1* in adult LMS. **a** Structural variant plots of all fusion transcripts involving *TP53* and *RB1* detected in 37 tumors. **b** Interchromosomal rearrangement resulting in a non-functional *TP53-TCERG1* fusion transcript in case LMS44 (top) and intrachromosomal rearrangement resulting in a non-functional *RB1-ATP8A2* fusion transcript in case LMS45 (bottom). TS transcriptome sequencing, chr chromosome. **c** Schematic representation of different genetic lesions targeting *TP53* and *RB1*. e exon, i intron, chr chromosome, UTR untranslated region

exists. Our findings not only advance current insights into the molecular basis of LMS, which are primarily based on lower-resolution microarray analyses and targeted sequencing of selected cancer genes, but may also have tangible clinical implications.

We observed that widespread genomic imbalances, in particular chromosomal losses affecting tumor suppressor genes such as *TP53*, *RB1*, and *PTEN*, are a hallmark of LMS, in keeping with previous studies[3, 43, 44]. However, an unexpected finding in this study was the very high frequency of biallelic *TP53* and *RB1* inactivation in LMS tumors. While it has been known that patients with Li-Fraumeni syndrome or hereditary retinoblastoma, which are associated with germline defects in *TP53* and *RB1*, respectively, have an increased risk for developing LMS as secondary malignancy[45, 46], the frequency of *TP53* and *RB1* disruption in sporadic LMS was reported to be in the range of 50% or lower[3, 43, 44]. In our study, whole-exome and transcriptome sequencing enabled the discovery that *TP53* and *RB1* are targeted by diverse genetic mechanisms (SNVs, indels, CNAs, chromosomal rearrangements, and microalterations (e.g. novel deletions affecting the *TP53* transcription start site)) in more than 90% of cases, establishing biallelic *TP53* and *RB1* inactivation as unifying feature of LMS development. In addition to providing an exhaustive picture of the tumor suppressor landscape of LMS, our data identify chromothripsis and WGD, crucial events in the pathogenesis of various cancers[47, 48], as previously unrecognized manifestations of genomic instability in this disease.

Our findings indicate that LMS cells primarily rely on ALT to overcome replicative mortality. However, the high prevalence of ALT in LMS (78% in our cohort) cannot be explained by the frequency of potentially deleterious *ATRX* alterations observed by us and others (49% and 16–26% of cases, respectively)[4, 49, 50]. In conjunction with the continuously growing list of putative telomere maintenance genes[51], our comprehensive catalog of genomic and transcriptomic alterations in LMS tumors provides an opportunity to select novel candidate drivers of ALT, such as *RBL2* and *SP100*, for future functional and mechanistic investigations.

Treatment of advanced-stage soft-tissue sarcoma, including LMS, is difficult, and for more than 30 years, doxorubicin, ifosfamide, and dacarbazine were the only active drugs in this setting. Additional agents have been tested, including gemcitabine, taxanes, trabectedin, pazopanib, and eribulin. However, none has proven superior to doxorubicin, and molecularly guided therapeutic strategies remain elusive[52]. Very recent data indicate that the anti-PDGFRA antibody olaratumab in combination with doxorubicin may improve survival, but these results await confirmation from phase 3 clinical trials[53]. We have found that most LMS tumors exhibit genomic "scarring" suggestive of impaired HRR of DNA double-strand breaks, which might represent a suitable target for therapeutic intervention through repositioning of small-molecule PARP inhibitors[38]. Given that the concept of "BRCAness" was primarily introduced in BRCA1/2-deficient epithelial cancers, further mechanistic evaluation of the HRR pathway and, most importantly, genomics-guided clinical trials in LMS patients will be necessary to formally establish whether a "BRCAness" phenotype confers sensitivity to these drugs as in breast, ovarian, and prostate cancer. However, preclinical observations[54] as well as preliminary data from a phase 1b trial of olaparib and trabectedin in unselected patients with relapsed bone and soft-tissue sarcomas (Grignani et al., ASCO Annual Meeting, 2016) suggest that this might be the case.

Apart from defective HRR, our analysis revealed additional leads for investigations into genetic alterations or deregulated cellular processes that might be exploited for therapeutic benefit.

For example, blockade of the PI3K-AKT-mTOR axis might be effective in LMS tumors (57% in our cohort) harboring *PTEN* alterations[55, 56]. Furthermore, it has been shown that ALT renders cancer cells sensitive to ATR inhibitors[57]. Amplifications of *TOP3A* (28%), *BLM* (12%), and *DNMT1* (12%) may provide a basis for the combinatorial use of the respective inhibitors with

chemotherapeutics or other targeted agents[58–60]. A recent study reported that the BLM DNA helicase drives an aggravated ALT phenotype in the absence of FANCD2 and FANCA[61], suggesting that BLM inhibition[59] may provide a means to target FANCD2- and FANCA-deficient LMS cells. Finally, DNA methyltransferase inhibitors enhance the cytotoxic effect of PARP inhibition in

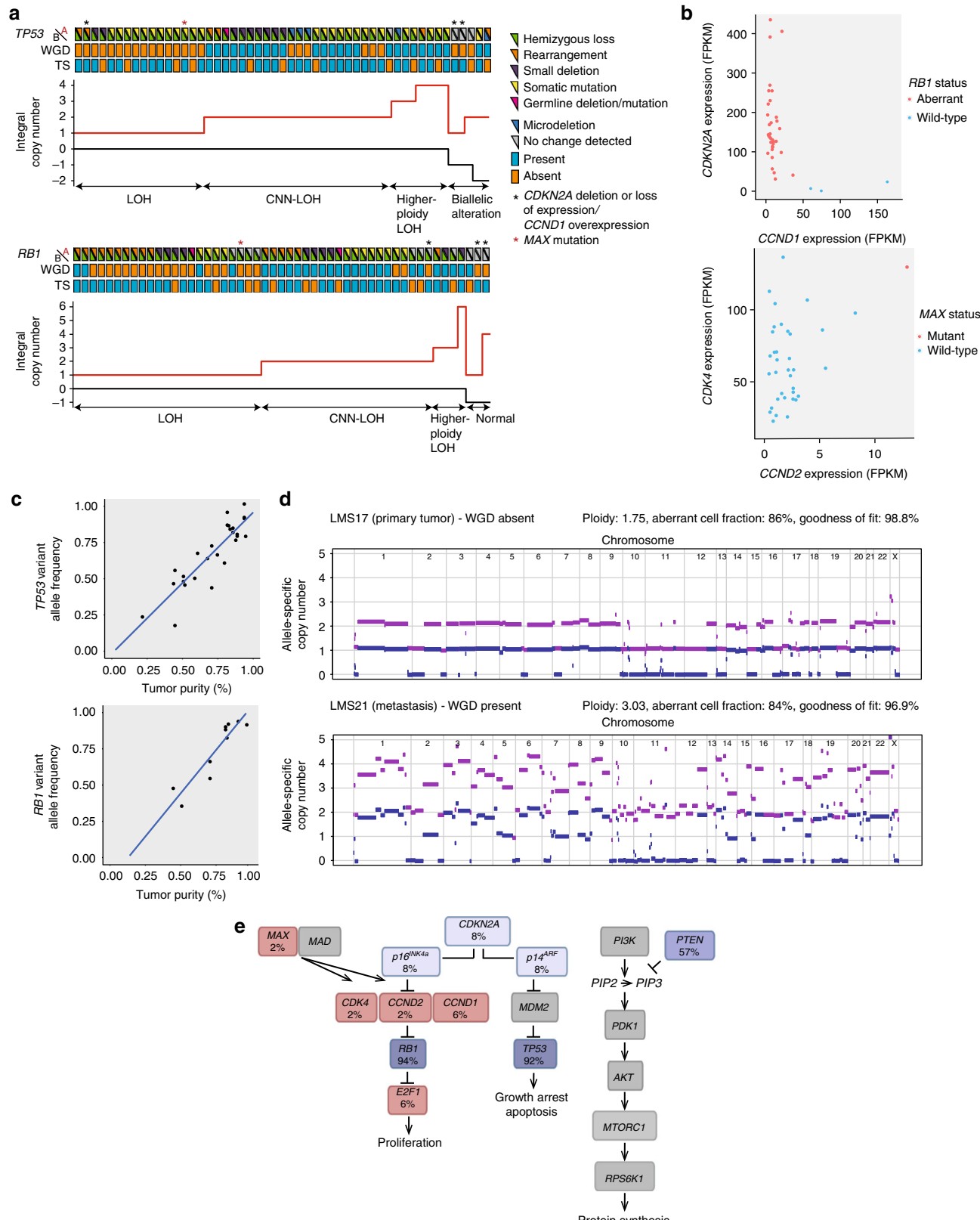

cancer cells[62], suggesting a mechanism-based strategy for combination therapy of LMS tumors with BRCAness, a subset of which are characterized by *DNMT1* copy number gains that may increase PARP binding to chromatin.

In summary, this comprehensive genomic and transcriptomic analysis has unveiled that LMS is characterized by substantial mutational heterogeneity, genomic instability, universal inactivation of TP53 and RB1, and frequent WGD. Furthermore, we have established that most LMS tumors rely on ALT to escape replicative senescence, and identified recurrent alterations in a broad spectrum of telomere maintenance genes. Finally, our findings uncover "BRCAness" as potentially actionable feature of LMS tumors, and provide a rich resource for guiding future investigations into the mechanisms underlying LMS development and the design of novel therapeutic strategies.

## Methods

**Patient samples.** For whole-exome and transcriptome sequencing, fresh-frozen tumor specimens and matched normal control samples (Supplementary Data 1) were collected from 49 adult patients who had been diagnosed with LMS according to World Health Organization criteria at four German cancer centers (NCT Heidelberg and Heidelberg University Hospital, Heidelberg; Mannheim University Medical Center, Mannheim; West German Cancer Center, Essen; Eberhard Karls University Hospital, Tübingen). Specimens were obtained from different anatomic sites, and the cohort included both treatment-naïve and previously treated patients (Supplementary Data 1). Samples were pseudonymized, and tumor histology and cellularity were assessed at the Institute of Pathology, Heidelberg University Hospital, prior to further processing. Twelve cases were excluded from transcriptome sequencing due to insufficient quantity and/or quality of RNA. Patient samples were obtained under protocol S-206/2011, approved by the Ethics Committee of Heidelberg University, with written informed consent from all human participants. This study was conducted in accordance with the Declaration of Helsinki.

**Cell lines.** SK-UT-1, SK-UT-1B, and MES-SA cells were purchased from American Type Culture Collection. SK-LMS-1 cells were provided by Sebastian Bauer (West German Cancer Center, Essen). Cell line identity and purity were verified using the Multiplex Cell Authentication and Contamination Tests (Multiplexion). All cell lines were regularly tested for mycoplasma contamination using the Venor GeM Mycoplasma Detection Kit (Minerva). Cell lines were cultured as follows: SK-LMS-1 in RPMI-1640 (Life Technologies), 15% FBS; SK-UT-1 and SK-UT-1B in MEM (Life Technologies), 10% FBS; MES-SA in McCoy's medium, 10% FBS. All media were supplemented with 1% penicillin/streptomycin and 1% L-glutamine (Biochrom).

**Isolation of analytes.** DNA and RNA from tumor specimens and DNA from control samples were isolated at the central DKFZ-HIPO Sample Processing Laboratory using the AllPrep DNA/RNA/Protein Mini Kit (Qiagen), followed by quality control and quantification using a Qubit 2.0 Fluorometer (Invitrogen) and a 2100 Bioanalyzer system (Agilent).

**Whole-exome sequencing.** Exome capturing was performed using SureSelect Human All Exon V5+UTRs in-solution capture reagents (Agilent). Briefly, 1.5 µg genomic DNA were fragmented to 150–200 bp insert size with a Covaris S2 device, and 250 ng of Illumina adapter-containing libraries were hybridized with exome baits at 65 °C for 16 h. Paired-end sequencing (2×101 bp) was carried out with a HiSeq 2500 instrument (Illumina).

**Mapping of whole-exome sequencing data.** Reads were mapped to the 1000 Genomes Phase 2 assembly of the Genome Reference Consortium human genome (build 37, version hs37d5) using BWA (version 0.6.2) with default parameters and maximum insert size set to 1000 bp. BAM files were sorted with SAMtools (version 0.1.19), and duplicates were marked with Picard tools (version 1.90). Sequencing coverages and additional quality parameters are summarized in Supplementary Data 1.

**Whole-genome sequencing.** Whole-genome sequencing libraries were prepared using the TrueSeq Nano Library Preparation Kit (Illumina) using the manufacturer's instructions. Paired-end sequencing (2×151 bp) was carried out with a HiSeq X instrument (Illumina).

**Mapping of whole-genome sequencing data.** Reads were mapped to the 1000 Genomes Phase 2 assembly of the Genome Reference Consortium human genome (build 37, version hs37d5) using BWA mem (version 0.7.8) with option -T 0. BAM files were sorted with SAMtools (version 0.1.19)[63], and duplicates were marked with Picard tools (version 1.125) using default parameters.

**Transcriptome sequencing.** RNA sequencing libraries were prepared using the TruSeq RNA Sample Preparation Kit v2 (Illumina), normalized to 10 nM, pooled to 11-plexes, and clustered on a cBot system (Illumina) to a final concentration of 10 pM with a spike-in of 1% PhiX Control v3 (Illumina). Paired-end sequencing (2×101 bp) was carried out with a HiSeq 2000 instrument (Illumina).

**Mapping of transcriptome sequencing data.** RNA sequencing reads were mapped with STAR (version 2.3.0e)[64]. For building the index, the 1000 Genomes reference sequence with GENCODE version 17 transcript annotations was used. For alignment, the following parameters were used: alignIntronMax 500,000, alignMatesGapMax 500,000, outSAMunmapped Within, outFilterMultimapNmax 1, outFilterMismatchNmax 3, outFilterMismatchNoverLmax 0.3, sjdbOverhang 50, chimSegmentMin 15, chimScoreMin 1, chimScoreJunctionNonGTAG 0, chimJunctionOverhangMin 15. The output was converted to sorted BAM files with SAMtools, and duplicates were marked with Picard tools (version 1.90).

**Detection of SNVs and small indels.** Somatic SNVs were detected from matched tumor/normal pairs with our in-house analysis pipeline based on SAMtools mpileup and bcftools with parameter adjustments and using heuristic filtering as previously described[65]. In brief, SAMtools (version 0.1.19) mpileup was called on the tumor BAM file with parameters RE -q 20 -ug to consider only reads with a minimum mapping quality of 20 and bases with a minimum base quality of 13. The output was piped to BCFtools (version 0.1.19) view, which, by using parameters -vcgN -p 2.0, reports all positions containing at least one high-quality non-reference base. From these initial SNV calls, the ones with at least five variant reads and a variant allele frequency of at least 5% were retained. Any variant call that was supported by reads from only one strand was discarded if one of the Illumina-specific error profiles occurred in a sequence context of ±10 bases around the SNV. For categorizing variants as germline or somatic, a pileup of the bases in the matched control sample was generated for each SNV position by SAMtools mpileup with parameters -Q 0 -q 1, considering uniquely mapping reads and not putting a restriction to base quality. For high-confidence somatic SNVs, the coverage at the position in the control must be at least ten, and less than 1/30 of the control bases may support the variant observed in the tumor. Variants that were located in regions of low mappability or overlapped with entries of the

**Fig. 5** Biallelic inactivation of *TP53* and *RB1* and whole-genome duplication (WGD) in adult LMS. **a** Combined analysis of genetic lesions and allele-specific copy number showing frequent biallelic inactivation of *TP53* and *RB1*. In the top panels, samples are plotted from left to right based on their copy number composition, and genetic lesions specific for the A and B alleles as well as the presence or absence of WGD are annotated according to the color code. Asterisks indicate cases with either loss of *CDKN2A* expression in combination with *CCND1* overexpression or *MAX* mutation. In the bottom panels, allele-specific integral copy numbers are plotted. Cases with retention of a single allele are assigned to the loss-of-heterozygosity (LOH) group, cases with one or more alleles derived from the same parental allele are assigned to the copy number-neutral (CNN) or higher-ploidy LOH groups, and cases with different combinations of maternal and paternal alleles are assigned to the normal or biallelic alteration group, respectively. TS transcriptome sequencing. **b** Scatter plots showing expression of *CDKN2A* and *CCND1* in cases with wild-type and aberrant *RB1* (left) and expression of *CDK4* and *CCND2* in cases with wild-type and mutant *MAX* (right). FPKM fragments per kilobase of transcript per million mapped reads. **c** Scatter plots showing congruency of *TP53* and *RB1* variant allele frequencies with tumor purity as detected by allele-specific copy number analysis. **d** Allele-specific copy-number profiles for a primary tumor/metastasis pair showing absence of WGD in the primary tumor (top) and presence of WGD in the metastasis (bottom). Chromosomes are represented along the horizontal axis, copy numbers are indicated along the vertical axis. The purple line indicates the total allele-specific copy number. The blue line indicates the minor allele-specific copy number. **e** Genes involved in cell cycle regulation or PI3K-AKT-mTOR signaling recurrently affected by genetic alterations in LMS tumors. Blue and red boxes denote genes with inactivating and activating lesions, respectively. Percentage values indicate the collective frequencies of SNVs, indels, CNAs, fusions, microalterations, and aberrant expression affecting the respective genes

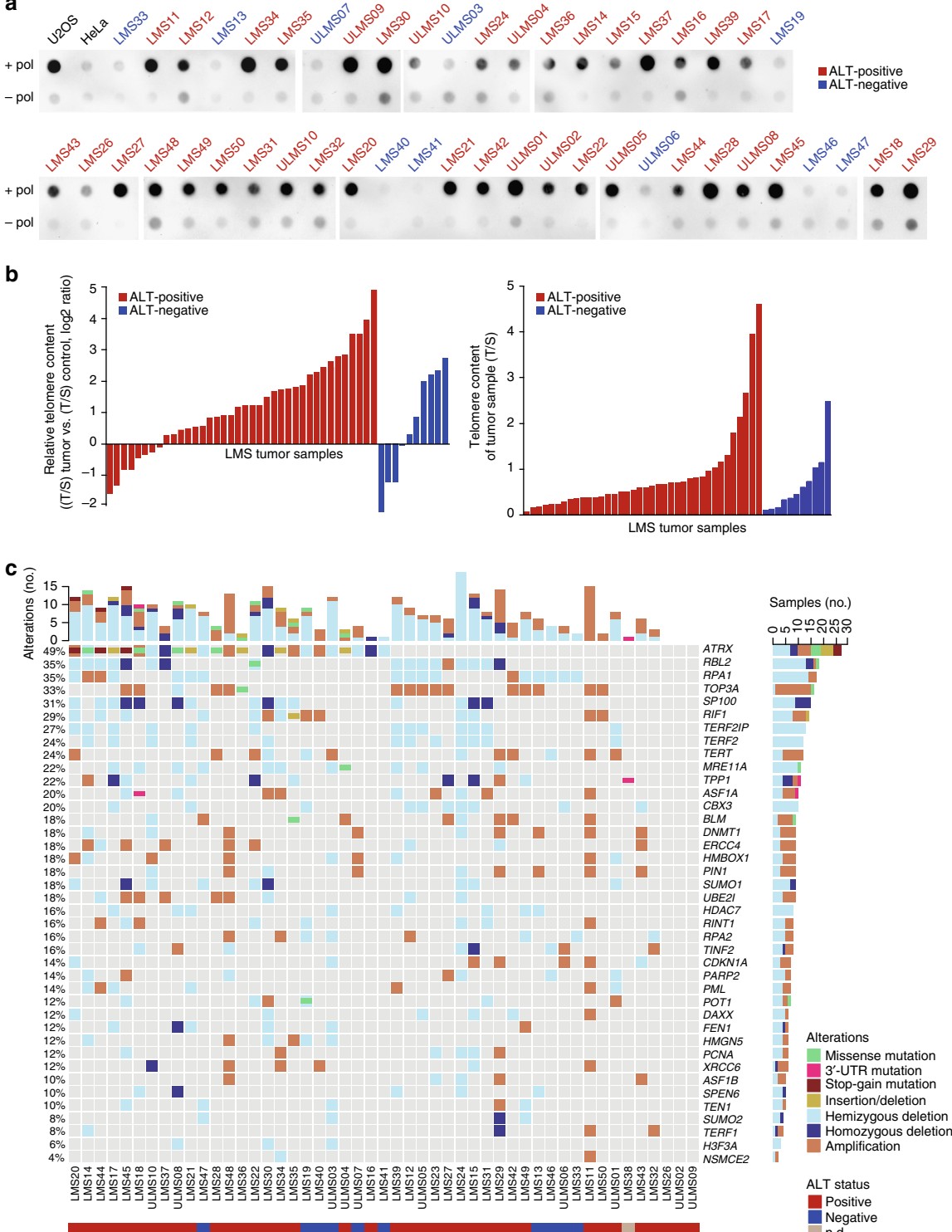

**Fig. 6** High frequency of alternative lengthening of telomeres (ALT) in adult LMS. **a** Detection of C-circles in LMS tumors and control cell lines (U2OS, positive control; HeLa, negative control). Shown are test samples (top row) and control samples (bottom row). ALT-positive samples, as inferred from the enriched C-circle signal, are indicated in red. ALT-negative samples are indicated in blue. +pol, with polymerase; −pol, without polymerase. **b** Measurement of telomere content in LMS tumors. Telomere quantitative PCR was performed on tumor and matched control samples, and telomere repeat signals were normalized to a single-copy gene (*36B4*; T/S ratio). Shown are the telomere contents of tumor samples relative to those of control samples (left) and the absolute telomere contents of tumor samples (right). **c** Recurrent alterations in telomerase maintenance genes in LMS tumors. Rows represent individual genes, columns represent individual tumors. Genes are sorted according to frequency of SNVs, indels, and CNAs (left). Bars depict the number of alterations for individual tumors (top) and genes (right). Types of alterations and ALT status are annotated according to the color codes (bottom). UTR untranslated region; n.d. not determined

hiSeqDepthTopPt1Pct track from the UCSC Genome Browser, Encode DAC Blacklisted Regions, or Duke Excluded Regions were excluded. High-confidence SNVs were also not allowed to overlap with any two of the following features at the same time: tandem repeats, simple repeats, low complexity, satellite repeats, or segmental duplications. After annotation with RefSeq (version September 2013)

using ANNOVAR, somatic, non-silent coding variants of high confidence were selected except for the analysis of mutational signatures, where all high confidence, including non-coding and silent, somatic variants were used. Small indels were identified by Platypus (version 0.5.2; parameters: genIndels = 1, genSNPs = 0, ploidy = 2, nIndividuals = 2) by providing matched tumor and control BAM files.

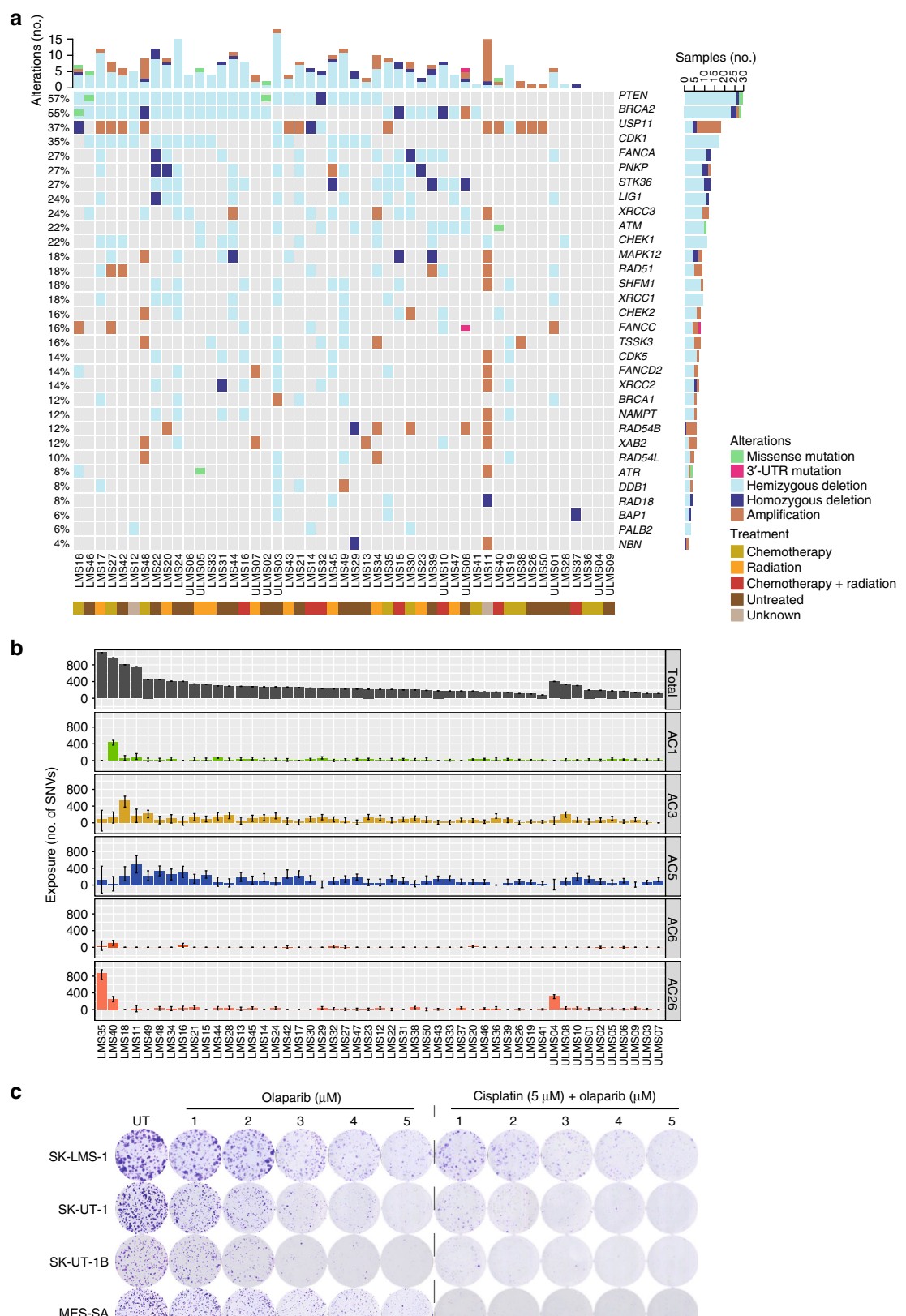

To be considered high confidence, somatic calls (control genotype 0/0) were required to either have the Platypus filter flag PASS or pass custom filters allowing for low variant frequency using a scoring scheme. Candidates with the badReads flag, alleleBias, or strandBias were discarded if the variant allele frequency was <10%. Additionally, combinations of Platypus non-PASS filter flags, bad quality values, low genotype quality, very low variant counts in the tumor, and presence of variant reads in the control were not tolerated. Indels were annotated with ANNOVAR, and somatic high-confidence indels falling into a coding sequence or splice site were extracted. SNVs and indels in LMS cell lines were called without matched control. In addition to filters and annotations described above, calls were further filtered for variants found in ExAC (version 0.3.1; http://exac.broadinstitute.org) with allele frequencies >0.0001, in the 1000 Genomes Phase 3 of the Genome Reference Consortium human genome with allele frequencies >0.001, and in our in-house control data set ($n = 655$) in more than 5% of the samples. Oncoprints integrating information on SNVs, indels, and CNAs were generated using the R package ComplexHeatmap[66].

**Supervised analysis of mutational signatures**. Using the package YAPSA (Yet Another Package for Signature Analysis)[67], a linear combination decomposition of the mutational catalog with predefined signatures from the COSMIC database (http://cancer.sanger.ac.uk/cosmic/signatures, downloaded in June 2016) was computed by non-negative least squares (NNLS). Prior to decomposition, the mutational catalog was corrected for different occurrences of the triplet motifs between the whole genome and the target capture regions used for whole-exome sequencing (function normalizeMotifs_otherRownames() from YAPSA). To increase specificity, the NNLS algorithm was applied twice; after the first execution, only those signatures whose exposures, i.e. contributions in the linear combination, were higher than a certain cut-off were kept, and the NNLS was run again with the reduced set of signatures. As the detectability of different signatures may vary, the following signature-specific cut-offs were determined in a random operator characteristic analysis using publicly available data on mutational catalogs of 7042 cancers (whole-genome sequencing, $n = 507$; whole-exome sequencing, 6535)[6] and mutational signatures from COSMIC: AC1, 0; AC2, 0.03404847; AC3, 0.139839; AC4, 0.02281439; AC5, 0; AC6, 0.003660315; AC7, 0.02841319; AC8, 0.1870989; AC9, 0.0953648; AC10, 0.0164065; AC11, 0.08238725; AC12, 0.1920715; AC13, 0.03769936; AC14, 0.03080224; AC15, 0.03182855; AC16, 0.3553548; AC17, 0.004075963; AC18, 0.2692715; AC19, 0.04038686; AC20, 0.05066134; AC21, 0.04219805; AC22, 0.03908793; AC23, 0.03900049; AC24, 0.04254174; AC25, 0.02448377; AC26, 0.02830282; AC27, 0.02223076; AC28, 0.0315642; AC29, 0.07392201; AC30, 0.06332517. The cut-offs are also stored in the R package YAPSA and can be retrieved with the following R code: library(YAPSA), data (cutoffs), cutoffCosmicValid_rel_df[6,]. Confidence intervals were computed using the concept of profile likelihoods. Likelihoods were computed from the distribution of the residues after NNLS decomposition (initial model of the data). To compute the confidence interval of a given signature, the exposure to this signature was perturbed and fixed as compared to the initial model, and the exposures to the remaining signatures computed again by NNLS, yielding an alternative model with one degree of freedom less. Likelihoods were again computed from the distribution of the residuals of the alternative model. Next, a likelihood ratio test for the log-likelihoods of the initial and alternative models was computed, yielding a test statistic and a $P$-value for the perturbation. To compute the limits of 95% confidence intervals, the perturbations corresponding to $P$-values of $0.05/2 = 0.025$ (two-sided likelihood ratio test) were computed by a Gauss–Newton method (R package pracma). The set of mutational signatures extracted from the LMS cohort was compared to the set of mutational signatures extracted from a background of 7042 cancer samples (whole-genome sequencing, $n = 507$; whole-exome sequencing, $n = 6535$)[6] by Fisher exact tests and subsequent correction for multiple comparisons according to the Benjamini–Hochberg method.

**Detection of germline variants**. For *TP53* and *RB1*, non-silent coding variants and splice-site mutations with read support in the matched normal control were filtered for single-nucleotide polymorphisms (SNPs) recorded in the 1000 Genomes Phase 2 assembly of the Genome Reference Consortium human genome with allele frequencies >0.001 or in ExAC (version 0.3.1) with allele frequencies >0.0001 and were visually inspected using Integrative Genomics Viewer to rule out sequencing artifacts.

**Identification of driver mutations**. Variant Call Format files were processed in combination with the whole-exome sequencing capture design BED file using an in-house pipeline that determines the recurrence of gene-specific mutations and scores the different possibilities of mutations per gene. Average gene expression levels were determined based on the previously calculated fragments per kilobase of transcript per million mapped reads values, which were calculated using Cufflinks[68]. The resulting files were used as input for MutSigCV[7] and processed using default parameters. $P$-values were corrected for multiple hypothesis testing using the Benjamini–Hochberg procedure, and genes with $q < 0.01$ were considered significantly mutated.

**Identification of significantly mutated gene networks**. Network analysis was performed using HotNet2 (version 1.0.1)[9], and the global interaction network (HINT+HI2012) was retrieved from the HotNet2 website (http://compbio-research.cs.brown.edu/pancancer/hotnet2). For each node (gene) in the global network, the −log10 $P$-value from MutSigCV served as the initial heat, which diffuses to adjacent nodes through edges (known interactions) with a weight $\delta > 0.008450441$. Areas accumulating more heat were identified as subnetworks, and significantly mutated subnetworks were determined based on a two-stage multiple hypothesis test[69] and 100 permutations of the global interaction network. Significant subnetworks ($P < 0.05$) were visualized with Cytoscape (version 2.6.2).

**Detection of DNA CNAs**. For LMS patient samples, copy numbers were estimated from exome data using read-depth plots and an in-house pipeline using VarScan2 copynumber and copyCaller modules. Regions were filtered for unmappable genomic stretches, merged by requiring at least 70 markers per called copy number event, and annotated with RefSeq genes using BEDTools. High-resolution CNA profiles were generated with CNVsvd (manuscript in preparation), which determines the total number of fragments from non-overlapping 250-bp windows based on the whole-exome sequencing capture design. Systematic variance introduced by sequence context or sequencing technology bias was captured through analysis of a reference data set, i.e. all normal controls with sufficient quality statistics, and these estimated local variance components were subsequently used to attenuate systematic variance in all sequenced specimens, including controls. Finally, normalized fragment count statistics were used to estimate CNA profiles. Segmentation was performed with PSCBS[70], segmentation files and windows used for CNA estimation were converted to a compound segmentation file and marker files that were used as input for GISTIC2.0[13], and processing was performed with default parameters. For LMS cell lines, copy numbers were estimated from whole-genome sequencing data using allele-specific copy number estimation from sequencing (ACEseq, manuscript in preparation), which employs tumor coverage and BAF and also estimates tumor cell content and ploidy. Allele frequencies were obtained during pre-processing of whole-genome sequencing data for all SNPs recorded in dbSNP (build 135), and positions with BAF values between 0.1 and 0.9 in the tumor were assumed to be heterozygous in the germline. To improve sensitivity for the detection of allelic imbalances, heterozygous and homozygous SNPs were phased with IMPUTE (version 2)[71]. In addition, the coverage for 10-kilobase (kb) windows with sufficient mapping quality and read density in an in-house control was recorded for the tumor and corrected for GC content- and replication timing-dependent coverage bias. The genome was segmented using the R package PSCBS[70], and segments were clustered according to coverage ratios and BAF values using $k$-means clustering. The R package mclust was used to determine the optimal number of clusters based on the Bayesian information criterion. Small segments (<9 kb) were attached to the more similar neighbor. Finally, tumor cell content and ploidy of a sample were estimated by fitting different tumor cell content and ploidy combinations to the data. Segments with balanced BAF values were fitted to even-numbered copy number states, whereas unbalanced segments could also be fitted to uneven copy numbers. Finally, estimated tumor cell content and ploidy values were used to compute the total and allele-specific copy number for each segment.

**Analysis of allele-specific copy number and tumor purity**. Allele-specific copy number profiles and tumor purity of LMS patient samples were analyzed with ASCAT[72] and Sequenza[73]. Input files for ASCAT were generated using an in-house algorithm that extracts fragment counts from tumor and matched normal BAM files at positions listed in dbSNP (build 137), and only sufficiently covered regions with >10 fragments and fragments with an alignment score >30 were considered. For further analysis, SNPs heterozygous in normal samples were used, and allele-specific copy number profiles for matched tumor samples were determined with standard parameters. For Sequenza, standard guidelines as specified in the reference manual were used. In the majority of cases, allele-specific copy numbers and tumor purity estimates were nearly congruent between ASCAT and Sequenza. For

**Fig. 7** Evidence for BRCAness in adult LMS. **a** Alterations in genes reported as synthetic lethal to PARP inhibition. Rows represent individual genes, columns represent individual tumors. Genes are sorted according to frequency of SNVs, indels, and CNAs (left). Bars depict the number of alterations for individual tumors (top) and genes (right). Types of alterations and treatment history are annotated according to the color codes (bottom). UTR untranslated region. **b** Contribution of mutational signatures to the overall mutational load in LMS tumors. Each bar represents the number of SNVs explained by the respective mutational signature in an individual tumor. Error bars represent 95% confidence intervals. AC Alexandrov-COSMIC. **c** Clonogenic assays showing dose-dependent sensitivity of LMS cell lines to continuous olaparib treatment (1–5 μM) with or without prior exposure to a 2-h pulse of cisplatin (5 μM). UT untreated

the remaining cases, optimal allele-specific copy number profiles were selected on the basis of tumor purity estimates provided by the pathologist and by comparing tumor purity estimates to the mutations with the most dominant variant allele frequencies, which frequently included alterations in *TP53*. WGD was determined by taking into account the estimated ploidy and the presence of two or multiple copies of most parental-specific chromosomes.

**Detection of chromothripsis**. Copy number state profiling of LMS tumors based on exome data to detect the alternating copy number states that are characteristic for chromothripsis was performed with the R package cn.MOPS (version 1.20.0)[74] using several exome-specific functions and modifications. The getSegmentReadCountFromBAM function was used in paired mode within enrichment of kit-specific target regions, and only properly paired reads with a mapping quality ≥20 were used for counting and duplicate reads removed. Tumors and control samples were compared using the referencecn.mops function adjusted with parameters from the exomecn.mops function and using the DNAcopy algorithm for segmentation. In addition, the obtained referencecn.mops log2 ratios were corrected to account for whole-chromosome gains and losses by shifting the ratio according to the proportion of total reads per chromosome related to the normal sample. Copy number plots with corrected log2 ratios were used for chromothripsis inference based on previously described criteria[75]. The rationale for adding a more conservative cut-off was that the number of copy number switches should be considered in relation to the size of the affected region, as it is more likely to observe ten copy number switches by chance on chromosome 1 than on chromosome 22 due to the size difference. Briefly, copy number plots were evaluated by counting switches between copy number states per chromosome independently by two researchers. The main criterion for calling chromothripsis was that the ratio of the number of alternating copy number state switches and the length of the affected region on an individual chromosome in Mb was higher than 0.2, which corresponds to at least ten alternating switches within 50 Mb. In this study, the minimum number of switches required for calling chromothripsis was set to 6, which should then have occurred within 30 Mb or less to satisfy the ≥0.2 cut-off.

**Analysis of transcriptome sequencing data**. After mapping of transcriptome data as described above, expression levels were determined per gene and sample as RPKM using RefSeq as gene model. For each gene, overlapping annotated exons from all transcript variants were merged into non-redundant exon units with a custom Perl script. Non-duplicate reads with mapping quality >0 were counted for all exon units with coverageBed from the BEDtools package[76]. Read counts were summarized per gene and divided by the combined length of its exon units (in kb), and the total number of reads (in millions) was counted by coverageBed. HTSeq-count (version 0.6.0)[77] was used to generate read count data at the exon level using a minimum mapping score of 1 and intersection non-empty mode and GENCODE version 17 as gene model. Size factor and dispersion estimation were calculated for raw count data before performing Wald statistics using DESeq2[78]. Regularized logarithm transformation was used for visualization and clustering of read count data. Unsupervised hierarchical clustering was performed using the 100 most variable genes. Normalized read count values for individual genes were centered and scaled (z-score), and quantile discretization was performed. Complete-linkage analysis with Euclidean distance measure was used for clustering. The heatmap was generated using the R package pheatmap. Principal component analysis was performed using singular value decomposition (prcomp) on the 1000 most variable genes to examine the co-variances between samples. Somatic SNVs and indels were annotated with RNA information by generating a pileup of the RNA BAM file using SAMtools. Variants were considered expressed if they were present in at least one high-quality RNA read. Fusion transcripts were determined using the TopHat2 post-alignment pipeline[79], and candidates with a score >300 were selected for further analysis. Circos plots were drawn with the R package OmicCircos.

**Validation of fusion transcripts**. Fusion transcripts were validated using RT-PCR and Sanger sequencing. RNA was reverse-transcribed using the High-Capacity cDNA Reverse Transcription Kit (Applied Biosystems). Breakpoint-spanning primers were designed manually and examined for secondary structures using mfold and off-target binding using Primer-BLAST. Melting temperatures of primers were calculated using the thermodynamic parameters of SantaLucia. Amplifications were carried out using *Taq* DNA Polymerase (Qiagen) according to the manufacturer's instructions. PCR products were visualized in 1% agarose gels and purified using the QIAquick PCR Purification Kit or the QIAquick Gel Extraction Kit (Qiagen). Direct sequencing was performed with the forward or reverse primer of the respective amplification.

**C-circle analysis**. C-circle analysis was performed as described previously[29]. Briefly, 30 ng genomic DNA from tumor samples was incubated with 1 x Φ29 Buffer, 0.2 mg/ml bovine serum albumin (BSA), 0.1% (v/v) Tween 20, 1 mM of each deoxyadenosine triphosphate (dATP), deoxyguanosine triphosphate (dGTP) and thymidine triphosphate (dTTP) , and with or without 7.5 U Φ29 DNA polymerase for 8 h at 30 °C, followed by inactivation for 20 min at 65 °C. After addition of 2x SSC, the DNA was dot-blotted with a 96-well dot blotter (Bio-Rad) onto a nylon membrane (Carl Roth), which was dried immediately and baked for 20 min at 120 °C. Hybridization, wash steps, and development were performed using the

TeloTAGGG Telomere Length Assay Kit (Roche) according to the manufacturer's instructions. Chemiluminescent signals of amplified C-circles were detected with a ChemiDoc MP Imaging System (Bio-Rad). Non-saturated exposures were used for evaluation, and tumor samples were classified as ALT-positive when the signal intensity of the complete reaction was at least twofold higher than that of the control without polymerase and at least threefold higher than the background intensity.

**Telomere quantitative PCR**. Telomere quantitative PCR was performed as described previously[80]. Briefly, 10 ng DNA from tumor or control samples was added to 1 μl LightCycler 480 SYBR Green I Master mix (Roche) and 500 nM of each forward and reverse primer in a 10 μl reaction. Primer sequences were as follows: Telomere forward: 5′-CGG TTT GTT TGG GTT TGG GTT TGG GTT TGG GTT TGG GTT-3′; Telomere reverse: 5′-GGC TTG CCT TAC CCT TAC CCT TAC CCT TAC CCT TAC CCT-3′; 36B4 forward: 5′-AGC AAG TGG GAA GGT GTA ATC C-3′; 36B4 reverse: 5′-CCC ATT CTA TCA TCA ACG GGT ACA A-3′. PCR conditions were as follows: 10 min at 95 °C, 40 cycles of 15 s at 95 °C and 60 s at 60 °C. For each tumor and control sample, a T/S ratio (telomere repeat signals normalized to a single copy gene (*36B4*)) was determined, and the T/S ratios of tumor samples were divided by those of matched control samples. The calculated log2 ratios represent the increase or decrease in telomere content in the tumor sample compared to the control sample.

**Clonogenic assays**. LMS cell lines (SK-LMS-1, SK-UT-1, MES-SA, 1×10³; SK-UT-1B, 2×10³) were seeded in six-well plates, and treatment with dimethyl sulfoxide (DMSO) or olaparib (1–5 μM; Selleck) was initiated 24 h after seeding and continued for 10 days, with drug replenishment and medium change every 2 days. Pretreatment with cisplatin (5 μM; Selleck) was performed for 2 h. Thereafter, cells were washed with phosphate buffered saline (PBS) and incubated with DMSO or olaparib as described above. Following drug treatment, cells were fixed with chilled methanol for 10 min, stained with 0.5% crystal violet in 25% methanol for 15 min, and photographed after overnight drying.

**Data availability**. Sequencing data were deposited in the European Genome-phenome Archive under accession EGAS00001002437.

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

## Acknowledgements

The authors thank N. Paramasivam, J. Park, the DKFZ-HIPO and NCT Precision Oncology Program (POP) Sample Processing Laboratory, the DKFZ Genomics and Proteomics Core Facility, and the DKFZ-HIPO Data Management Group for technical support. We also thank K. Beck, D. Richter, and P. Lichter for infrastructure and program development within DKFZ-HIPO and NCT POP and D. Braun for providing the TelNet gene list. Tissue samples were provided by the NCT Heidelberg Tissue Bank in accordance with its regulations and after approval by the Ethics Committee of Heidelberg University. This work was supported by grants H018, H021, and H028 from DKFZ-HIPO and NCT POP, as well as by the e:Med Systems Medicine Program of the German Federal Ministry of Education and Research within the CancerTelSys Consortium (grant 01ZX1302). M.A.S. is the recipient of a Rubicon Fellowship from Nederlandse Organisatie voor Wetenschappelijk Onderzoek (grant 019.153LW.038). D.H. is a member of the Hartmut Hoffmann-Berling International Graduate School of Molecular and Cellular Biology and of the MD/PhD Program of Heidelberg University. C.S. was supported by an Emmy Noether Fellowship from the German Research Foundation.

## Author contributions

P.C., I.C., K.I.D., and S.R. designed and performed experiments. P.C., S.S.M., M.A.S., D.H., S.-H.W., S.R., M.H., A.E., K.K., L.S., B.Kl., B.B., B.H., and M.R. analyzed and interpreted bioinformatics data. B.Ka., C.E.H., G.E., H.G., S.G., H.-G.K., G.O., B.L., S.B., S.S., A.U., G.M., M.R., P.H., and S.F. contributed patient samples. A.S., W.W., and G.M. performed pathology review. M.Z., M.S., R.E., E.S., R.M.H., S.W., C.v.K., H.G., S.G., K.R., B.B., and M.R. provided essential reagents, expertise, and infrastructure. P.C., S.S.M., M.A.S., D.H., I.C., C.S., and S.F. wrote the manuscript, which was reviewed and edited by all co-authors. C.S. and S.F. conceived and supervised the project.

## Additional information

**Competing interests:** The authors declare no competing financial interests.

Priya Chudasama[1], Sadaf S. Mughal[2,3], Mathijs A. Sanders[4,31,31], Daniel Hübschmann[5,6,7], Inn Chung[8], Katharina I. Deeg[8], Siao-Han Wong[2], Sophie Rabe[1], Mario Hlevnjak[9], Marc Zapatka[9], Aurélie Ernst[9,10], Kortine Kleinheinz[5], Matthias Schlesner[5], Lina Sieverling[2], Barbara Klink[11,12,13], Evelin Schröck[11,12,13], Remco M. Hoogenboezem[4], Bernd Kasper[14], Christoph E. Heilig[15], Gerlinde Egerer[15], Stephan Wolf[16], Christof von Kalle[1,10,17,18], Roland Eils[5,6,18], Albrecht Stenzinger[10,19], Wilko Weichert[20,21], Hanno Glimm[1,10,17], Stefan Gröschel[1,10,15,17,22], Hans-Georg Kopp[23,24], Georg Omlor[25], Burkhard Lehner[25], Sebastian Bauer[26,27], Simon Schimmack[28], Alexis Ulrich[28], Gunhild Mechtersheimer[19], Karsten Rippe[8], Benedikt Brors[2,10], Barbara Hutter[2], Marcus Renner[19], Peter Hohenberger[14,29], Claudia Scholl[1,10,30] & Stefan Fröhling[1,10,17]

[1]Division of Translational Oncology, National Center for Tumor Diseases (NCT) Heidelberg and German Cancer Research Center (DKFZ), 69120 Heidelberg, Germany. [2]Division of Applied Bioinformatics DKFZ and NCT Heidelberg, 69120 Heidelberg, Germany. [3]Faculty of Biosciences, Heidelberg University, 69120 Heidelberg, Germany. [4]Department of Hematology, Erasmus Medical Center, 3015 CN Rotterdam, The Netherlands. [5]Division of Theoretical Bioinformatics DKFZ, 69120 Heidelberg, Germany. [6]Department of Bioinformatics and Functional Genomics, Institute of Pharmacy and Molecular Biotechnology, Heidelberg University and BioQuant Center, 69120 Heidelberg, Germany. [7]Department of Pediatric Immunology, Hematology and Oncology, Heidelberg University Hospital, 69120 Heidelberg, Germany. [8]Research Group Genome Organization and Function, DKFZ and BioQuant Center, 69120 Heidelberg, Germany. [9]Division of Molecular Genetics DKFZ, 69120 Heidelberg, Germany. [10]German Cancer Consortium (DKTK), 69120 Heidelberg, Germany. [11]Institute for Clinical Genetics, Faculty of Medicine Carl Gustav Carus, Technical University Dresden, 01307 Dresden, Germany. [12]NCT Dresden, 01307 Dresden, Germany. [13]DKTK, 01307 Dresden, Germany. [14]Sarcoma Unit, Interdisciplinary Tumor Center Mannheim, Mannheim University Medical Center, Heidelberg University, 68167 Mannheim, Germany. [15]Department of Internal Medicine V, Heidelberg University Hospital, 69120 Heidelberg, Germany. [16]Genomics and Proteomics Core Facility DKFZ, 69120 Heidelberg, Germany. [17]Section for Personalized Oncology, Heidelberg University Hospital, 69120 Heidelberg, Germany. [18]DKFZ-Heidelberg Center for Personalized Oncology (HIPO), 69120 Heidelberg, Germany. [19]Institute of Pathology, Heidelberg University Hospital, 69120 Heidelberg, Germany. [20]Institute of Pathology, Technical University Munich, 81675 Munich, Germany. [21]DKTK, 81675 Munich, Germany. [22]Research Group Molecular Leukemogenesis DKFZ, 69120 Heidelberg, Germany. [23]Department of Hematology and Oncology, Eberhard Karls University, 72076 Tübingen, Germany. [24]DKTK, 72076 Tübingen, Germany. [25]Department of Orthopedics, Heidelberg University Hospital, 69118 Heidelberg, Germany. [26]Sarcoma Center Western German Cancer Center, 45147 Essen, Germany. [27]DKTK, 45147 Essen, Germany. [28]Department of General, Visceral and Transplantation Surgery, Heidelberg University Hospital, 69120 Heidelberg, Germany. [29]Department of Surgery, Mannheim University Medical Center, Heidelberg University, 68167 Mannheim, Germany. [30]Division of Applied Functional Genomics DKFZ, 69120 Heidelberg, Germany. [31]Present address: Wellcome Trust Sanger Institute, Hinxton, UK. Priya Chudasama, Sadaf S. Mughal, Mathijs A. Sanders and Daniel Hübschmann contributed equally to this work.

