## [Peer Review File · Nature Communications]

Reviewers' comments:

Reviewer #1 (Remarks to the Author):

This manuscript describes whole genome and gene expression analyses of a series of leiomyosarcoma. This seems the most comprehensive analyses to date for these tumors and will be useful data for future studies.

Comments

Unfortunately the conclusions are not very novel – the main conclusions of heterogeneity, TP53 and RB1 inactivation, alternative telomere lengthening (ALT) involvement and what the authors “BRCAness” are already in the literature (and as the authors quote, olaparib has very recently been tested in a combination study in a leiomyosarcoma cell line).

Evidence for chromothripsis is possibly novel with a high proportion of cases (35%)? This is not mentioned in the abstract and presented in the supplemental data.

The meaning of a “large” series in the abstract should be specified $n =$.

In the PCA analyses (although numbers are actually quite low), are the results the same considering the different locations of LMS (ovarian versus other) and local versus metastatic cases studied?

The aberrant RBL2, and SP100 etc, are associated and indicative of the ALT but do not provide mechanistic evidence.

Any direct evidence for homologous recombination repair is lacking and evidence for synthetic lethality is largely circumstantial (i.e. specific mutations expected to contribute in particular cell lines).

Data look good/high quality as is the writing but overall, the study is largely descriptive in nature.

Reviewer #2 (Remarks to the Author):

Comments on Chudasama et al.

The authors report the exome and transcriptome sequencing of a set of leiomyosarcomas. They find frequent mutations in several well known cancer genes and observe three distinct expression subsets. Although there are other similar studies published, this is the largest study to date integrating DNA and RNA analysis of this form of sarcoma. In addition, an effort is made to characterize the samples with respect to ALT status. While in general this work is done well and clearly presented, there are some questions for the authors mainly regarded to the challenging problem of interpreting genome complexity in cancers with such extensive abnormalities.

The authors should address the following points:

1) Fig 1. Network analysis: “The frequency of SNVs or indels served as the initial heat”
Does this mean that the input was the raw mutational data, not e.g. the derived MutSig values? In this case, # of SNVs should be at least corrected for e.g. gene length (see MutSig manuscript). The resulting network seems to include mainly genes that are unaffected by SNVs and/or structural variation and the signal stems purely from TP53/RB1. If none of the other network members is

affected by SNVs/structural variation, then the result seems more like an artifact than a sensible result. It would seem to be very non-specific for LMS as this type of mutation is present in other sarcoma types as well as some non-sarcomas. In the absence of better evidence, the network result has little novelty and probably should be removed or removed from the main text since it adds very little new information.

Fig S1 B. Are these examples from a single case or multiple cases? This is not clear from the legend.

Fig 2. A. What criteria were used to credential a gene as an "established cancer gene". Fig. 2C adds very little to panels A and B and should be removed or relegated to the supplement.

Transcriptome analysis: "Both principal component (PC) analysis (Fig. 3A) and unsupervised hierarchical clustering (Fig. 3B) revealed three distinct subgroups of patients"
But not the same number of members for each group. Also, why show the PCA in Fig 3a, it does not give more information than Fig 3b.

Figure 3b: color legend indicates what? Values go from 2-10, I guess gene expression, but not clear what unit. Also it is important that tumor purity is not driving signal (unlikely, but given that at least 8 tumors <50% pure it is possible and is straightforward to check). Also the strong signal for platelet degranulation, coagulation etc. suggests the possibility of non-tumor cell admixture. The genes driving that signature should be included in the supp. Table and checked for their expression in reference databases of normal cell types.

Fig 4. Too many IGV screen shots. Panels A,B,C suffice to make the main points. D could be in the supplement.

Fig 5 b/c. Remove background grid from the figures. Fig. 5 C/D legend is incorrect. Check text for correct call outs. Fig 5 E, What types of mutation are included in the % mutation figures (they don't match Fig 1C.) Inclusion of MAX: Really, MAX does not seem to be a major player, and it is not clear that it deserves emphasis at this low frequency (N of 1). Without more data to validate MAX the importance of this gene as a mutational target is uncertain.

ALT. Since the authors make a specific claim that this is the highest frequency of ALT observed, there needs to be more detail as to the justification of the cutoffs used. As the assays give a continuous output, there are always a number of samples falling near the cutoffs. Notably, normal blood and tumor arise from different lineages which need not have the same "normal" telomere repeat content. This can cause misinterpretation of the telomere content. The C-circle while reliable when strong is not necessarily easy to evaluate at low signals. Were there repeat assays for each tumor or only 1?. Also, while the approach taken to investigate telomere associated genes is interesting in an abstract sense, it is very hard to interpret Fig 6 C since these genes have diverse roles with varying degrees of mechanistic connection to telomere biology. In general, as CNAs are the principal drivers of this figure, this problem devolves to the challenging interpretation of the CNA plots in Fig 2A. Since a large portion of the genome has frequent CNAs, then any large set of genes (many hundreds are listed in TELNET) will generate positive results. This is a great way to generate hypotheses, but doesn't really prove much. Also, since ATRX is on X, copy number is gender biased, ATRX CNA/LOH may not always be meaningful. In addition, I am not sure that amplification deserves to be added into the relevant mutation list for this (or any other gene) whose loss is associated with ALT.

"BRCAness" Fig 7 A. Essentially the same problem applies to this plot as to Fig 6 C. This is a fine way to explore the data, but is insufficient to establish the point. Regarding 7B, please provide the selected YASPA cut-off values, as the analysis could not be reproduced with the provided data. Furthermore,

please provide significance values for the analysis. It is not clear if the pattern is random or statistically significant. In general, signature 3 does not seem to be the dominant pattern in most instances. The stacked barplots do not help. While it is absolutely true that signature 3 is associated in breast/ovarian cancers with impaired homologous recombination, the absolute strength of the signature in most other cancers is much weaker than in those cancer and may be indicative of other pathways with similar end effects. Similarly, while I am sure that the data showing some degree of sensitivity of cells to olaparib is correct, this does not always correlate with meaningful clinical sensitivity. While appealing to connect LMS instability to a drug target, it could be misleading to propose defective HR as a target since it is not firmly established by the available data. Delving into the many aspects of DNA damage and repair which could be disturbed in LMS would require much more analysis supported by mechanistic experimentation in model systems.

RE: NCOMMS-17-15211
Integrative genomic and transcriptomic analysis of leiomyosarcoma

We thank the Reviewers for their insightful and constructive comments, which have greatly improved this work. Please find enclosed a revised manuscript that has been modified in accordance with the Reviewers' recommendations. Our specific responses are detailed individually below.

Reviewer #1

This manuscript describes whole genome and gene expression analyses of a series of leiomyosarcoma. This seems the most comprehensive analyses to date for these tumors and will be useful data for future studies.

We are grateful for these favorable comments, and are delighted that the Reviewer found this an important study.

Unfortunately the conclusions are not very novel – the main conclusions of heterogeneity, *TP53* and *RB1* inactivation, alternative telomere lengthening (ALT) involvement and what the authors “BRCAness” are already in the literature (and as the authors quote, olaparib has very recently been tested in a combination study in a leiomyosarcoma cell line).

As pointed out by the Reviewer and acknowledged in our manuscript, alterations of *TP53* and *RB1* in patients with leiomyosarcoma (LMS) have been observed in previous studies, which were primarily based on lower-resolution microarray analyses and targeted sequencing of selected cancer genes. However, the frequency of *TP53* and *RB1* disruption in sporadic LMS was reported to be in the range of 50% or lower (Agaram et al. *Genes Chromosomes Cancer* 55:124-30, 2016; Ito et al. *Clin Cancer Res* 17:416-26, 2011; Perot et al. *Am J Pathol* 177:2080-90, 2010). In contrast, we have used whole-exome and RNA sequencing to discover that biallelic disruption of *TP53* and *RB1* is near-obligatory in LMS, establishing inactivation of these tumor suppressor genes as unifying feature of LMS development. Second, our study has revealed that *TP53* and *RB1* are targeted by a broad spectrum of genetic mechanisms, including single-nucleotide variants, small insertions/deletions, DNA copy number alterations, chromosomal rearrangements, and microalterations, some of which (e.g. deletions affecting the *TP53* transcription start site) have not been described previously. Third, we provide, for the first time, temporal evidence indicating that inactivation of *TP53* and *RB1* are truncal events in LMS pathogenesis that give rise to widespread genomic instability, including whole-genome duplication as previously unrecognized feature of this disease in 55% of cases.

Previous genetic studies have identified alterations of the chromatin remodeling factor ATRX, an established suppressor of alternative lengthening of telomeres (ALT), in LMS patients. In addition to DNA and RNA sequencing, we have employed the C-circle assay, which provides a quantitative measure of ALT through direct detection of telomere-derived, partially double-stranded circles of DNA (Oganesian

and Karlseder. *Mol Cell* 42:224-236, 2011), to identify for the first time that this telomere maintenance mechanism is active in nearly half of LMS patients with wild-type *ATR*X. Moreover, comparison of our comprehensive genomic and transcriptomic datasets with catalogs of putative telomere maintenance genes has allowed us to nominate novel candidate drivers of ALT, such as *RBL2* and *SP100*, that are affected by newly identified recurrent alterations in LMS tumors.

To our knowledge, it has not been reported that most LMS tumors are characterized by genomic imprints of defective homologous recombination DNA repair, a phenotype frequently referred to as “BRCAness” (Lord and Ashworth. *Nat Rev Cancer* 16:110-120, 2016), including multiple structural rearrangements, alterations of various DNA repair genes, and enrichment of specific mutational signatures associated with failure of double-strand DNA break repair that can only be detected by whole-exome/genome sequencing. In support of our hypothesis that defective homologous recombination might confer sensitivity to pharmacologic PARP inhibition as in common epithelial cancers, a recent preclinical study has shown that the growth of LMS cell lines was impaired by incubation with olaparib or veliparib; however, no attempt was made to associate this property with specific genetic characteristics (Pignochino et al. *Mol Cancer* 16:86, 2017). Similarly, an ongoing phase 1b trial in unselected patients with bone and soft-tissue sarcomas has demonstrated clinical activity of olaparib in combination with trabectedin in a subset of cases; however, information on genomic characteristics that might predict response to these agents is lacking (ClinicalTrials.gov Identifier NCT02398058; Grignani et al. American Society of Clinical Oncology Annual Meeting 2016).

Based on the above considerations, we hope the Reviewer will agree that our report provides significant novel information.

Evidence for chromothripsis is possibly novel with a high proportion of cases (35%)? This is not mentioned in the abstract and presented in the supplemental data.

We thank the Reviewer for rightly stressing the presence of chromothripsis in a remarkably high proportion of LMS cases. To address the Reviewer’s point, we have specified this finding in the abstract and moved the relevant figure from the supplementary data to Fig. 2c.

The meaning of a “large” series in the abstract should be specified n= .

We agree, and have amended the abstract as suggested.

In the PCA analyses (although numbers are actually quite low), are the results the same considering the different locations of LMS (ovarian versus other) and local versus metastatic cases studied?

To address this interesting question, we have evaluated the different subgroups identified by principal component analysis (PCA) for correlation with various clinical features (e.g. anatomic site of origin,

disease stage [primary tumor vs. local recurrence vs. metastasis], treatment history [treatment-naïve vs. previously treated]). As shown in Figure R1 below, the clustering pattern does not separate tumors from different anatomic regions (top panel) or samples reflecting different disease stages (bottom panel), pointing to tumor-intrinsic gene expression as the main driver of the admittedly low level of variation among LMS cases.

Figure R1

The aberrant *RBL2*, and *SP100* etc, are associated and indicative of the ALT but do not provide mechanistic evidence.

We share the Reviewer's interest in the mechanistic roles of newly identified candidate drivers of ALT such as *RBL2* and *SP100*. However, we would also like to emphasize that the scope of this study was to provide the first comprehensive, high-resolution genomic and transcriptomic map of LMS, which will serve as a valuable resource for guiding future mechanistic investigations as well as the design of novel therapeutic strategies.

Based on the evidence presented in our report, we are currently exploring the role of non-canonical, ATRX-independent activation of ALT in the pathogenesis of LMS, which potentially involves disruption of *RBL2* and *SP100* function; however, the lack of ALT-positive LMS cell lines has turned out to be a challenge. Furthermore, the induction ALT, a complex multi-step process, by RNA interference-mediated gene knockdown or CRISPR/Cas9-mediated gene knockout has not been successful for us and several other groups, as established cancer cell lines are either captured in replicative senescence or upregulate telomerase expression before they activate ALT. Our current efforts in the laboratory therefore include the establishment of LMS cell lines and xenografts from fresh patient material, which will hopefully also yield ALT-positive models that can be used, in the intermediate to long term, to study the mechanisms underlying the exceptionally high frequency of ALT in LMS tumors.

Any direct evidence for homologous recombination repair is lacking and evidence for synthetic lethality is largely circumstantial (i.e. specific mutations expected to contribute in particular cell lines).

We have subjected a cohort of primary human LMS samples as well as LMS cell lines to a comprehensive analysis of known genomic imprints of homologous recombination deficiency, i.e. specific patterns of structural rearrangements, alterations of individual genes known to be involved in homologous recombination, and enrichment of specific mutational signatures that have been proposed as surrogate marker of defective homology directed double-strand DNA break repair (Alexandrov et al. Nat Commun 6:8683, 2015; Lord and Ashworth. Nat Rev Cancer 16:110-120, 2016; Davies et al. Nat Med 23:517-525, 2017). Of note, this approach goes far beyond the genetic criteria that are usually applied in patients with epithelial cancers to determine their eligibility for PARP inhibitor treatment (see Mateo et al. N Engl J Med 373:1697-1708, 2015 for a recent example). Furthermore, we have shown that olaparib and cisplatin treatment impair the growth of LMS cell lines whose genomic profiles resemble those of primary LMS patient samples, including damaging alterations of multiple genes that have been associated with defective homologous recombination and induction of synthetic lethality to PARP inhibition in the recent literature. In our view, these data represent compelling evidence that the concept of "BRCAness" as clinically actionable genomic feature likely extends to LMS, and provide a rationale for genomics-guided clinical trials of pharmacologic PARP inhibition, either alone or in combination with DNA-damaging cytotoxic agents, in this intractable tumor entity. In this regard, we would like to note that our findings have spurred the development of a multi-institutional clinical trial evaluating the efficacy of pharmacologic PARP inhibition in combination with trabectedin in patients

with advanced-stage cancer, including LMS, and defective homologous recombination as assessed by comprehensive genomic profiling (ClinicalTrials.gov Identifier NCT03127215).

As alluded to by the Reviewer, we have not mechanistically evaluated the homologous recombination pathway, e.g. by directly assessing RAD51 foci formation via immunofluorescence or other DNA repair complexes via immunohistochemistry. Such experiments would be technically complex and difficult to standardize as they would require viable LMS samples to be exposed to DNA damage *ex vivo* before functional biomarkers of defective homologous recombination can be evaluated, and we hope the Reviewer will agree that this task is beyond the remit of the current study, which aimed to determine the genomic and transcriptomic landscape of LMS.

Data look good/high quality as is the writing but overall, the study is largely descriptive in nature.

We are grateful for these positive remarks. Despite being descriptive, integrative analyses of multi-omics data have provided invaluable insights into the molecular pathogenesis of many cancer entities, and have paved the way for novel, molecular mechanism-guided therapies. We believe that our study accomplishes the same for LMS, an aggressive malignancy whose molecular landscape remains poorly characterized, as the data have uncovered key biological features of LMS and provide a comprehensive framework for future studies that pertain to in-depth investigations into the mechanisms underlying LMS development as well as new approaches to clinical management.

Reviewer #2

The authors report the exome and transcriptome sequencing of a set of leiomyosarcomas. They find frequent mutations in several well-known cancer genes and observe three distinct expression subsets. Although there are other similar studies published, this is the largest study to date integrating DNA and RNA analysis of this form of sarcoma. In addition, an effort is made to characterize the samples with respect to ALT status. While in general this work is done well and clearly presented, there are some questions for the authors mainly regarded to the challenging problem of interpreting genome complexity in cancers with such extensive abnormalities.

We thank the Reviewer for these positive and encouraging comments, and appreciate the constructive feedback on several aspects of our study.

The authors should address the following points:

1) Fig 1. Network analysis: “The frequency of SNVs or indels served as the initial heat”

Does this mean that the input was the raw mutational data, not e.g. the derived MutSig values? In this case, # of SNVs should be at least corrected for e.g. gene length (see MutSig manuscript). The resulting network seems to include mainly genes that are unaffected by SNVs and/or structural variation and the signal stems purely from TP53/RB1. If none of the other network members is affected by SNVs/structural variation, then the result seems more like an artifact than a sensible result. It would seem to be very non-specific for LMS as this type of mutation is present in other sarcoma types as well as some non-sarcomas. In the absence of better evidence, the network result has little novelty and probably should be removed or removed from the main text since it adds very little new information.

We are grateful for these helpful suggestions. To address the Reviewer’s point, we have re-run the analysis using *P* values retrieved from MutSigCV, and as expected, *TP53* and *RB1* remain the “hottest” nodes. In contrast to our initial analysis, which was based on raw mutation frequencies, *TP53* and *RB1* now emerge as the “hubs” of two separate subnetworks encompassing additional genes related to MAPK signaling, regulation of muscle cell proliferation, and regulation of mRNA stability. These data have been included in the revised manuscript. Additional significantly mutated ($P < 0.05$) subnetworks comprising at least five genes are shown in Figure R2 below.

Reflecting the purpose of the HotNet2 algorithm – to find signaling and regulatory networks mutated more than expected by chance via identifying combinations of rare somatic mutations across pathways and protein complexes (Leiserson et al. Nat Genet 47:106-114, 2015) – most components of the networks shown in Fig. 1c of the revised manuscript and Figure R2 below are mutated at low frequencies, whereas wild-type genes are not included in networks identified by HotNet2.

We recognize that alterations of *TP53* and *RB1* are not specific for LMS. However, our study demonstrates for the first time that biallelic disruption of these genes is near-obligatory in LMS, whereas previous studies had reported frequencies of less than 50%. Since this newly identified role of *TP53* and

RB1 inactivation as unifying feature of LMS development is further substantiated by the finding of two dominant gene networks centered on *TP53* and *RB1*, we would suggest to leave the HotNet2 results in the main text and Fig. 1 of the revised manuscript, but are of course also happy to move them to the supplementary data if viewed of importance by the Reviewer.

Figure R2

Fig S1 B. Are these examples from a single case or multiple cases? This is not clear from the legend.

The wording of the figure caption (“Read-depth plots of case LMS24 …”) indicates that these examples are from a single case.

Fig 2. A. What criteria were used to credential a gene as an “established cancer gene”. Fig. 2C adds very little to panels A and B and should be removed or relegated to the supplement.

Our definition of “established cancer genes” was based on the Cancer Gene Census database (<http://cancer.sanger.ac.uk/census>; Futreal et al. Nat Rev Cancer 4:177-183, 2004 [Reference 11 of the manuscript]) developed at the Wellcome Trust Sanger Institute and regularly updated by the COSMIC (Catalogue Of Somatic Mutations In Cancer) curation team.

By displaying both recurrent focal gains and losses (Fig. 2b) as well as broad and focal alterations on a case-by-case basis (Fig. 2c), we intended to provide an overview of the entire spectrum of DNA copy number alterations in LMS and highlight the high degree of genomic instability in individual patients. However, we completely agree with the Reviewer that Fig. 2c adds little information, and have moved the GISTIC2.0 heatmap to the supplementary data as suggested.

Transcriptome analysis: “Both principal component (PC) analysis (Fig. 3A) and unsupervised hierarchical clustering (Fig. 3B) revealed three distinct subgroups of patients”

But not the same number of members for each group. Also, why show the PCA in Fig 3a, it does not give more information than Fig 3b.

We agree that principal component analysis (PCA) and unsupervised hierarchical clustering support very similar interpretations of the data, even though it is plausible, owing to differences in the underlying methodology, that PCA (which uses *k*-means clustering to partition the data) and unsupervised hierarchical clustering may result in differential grouping of samples with minor differences in gene expression. PCA also allowed us to test the stability of clusters when using larger ($n \geq 5,000$) gene sets. To avoid the redundancy Reviewer #2 pointed out correctly, we have moved Fig. 3a to the supplementary data.

Figure 3b: color legend indicates what? Values go from 2-10, I guess gene expression, but not clear what unit. Also it is important that tumor purity is not driving signal (unlikely, but given that at least 8 tumors <50% pure it is possible and is straightforward to check). Also the strong signal for platelet degranulation, coagulation etc. suggests the possibility of non-tumor cell admixture. The genes driving that signature should be included in the supp. Table and checked for their expression in reference databases of normal cell types.

We have amended Fig. 3b (Fig. 3a of the revised manuscript) to specify that the color code indicates normalized read count values for individual genes, which were centered, scaled (z-score), and quantile-discretized. We appreciate the Reviewer’s concern and can not rule out the possibility of contaminating RNA from adjacent tissues or infiltrating immune cells. However, based on Figure R1 (top panel) on Page 3 and Figure R3 below, we would argue that the separation of tumor samples is likely independent of their anatomic location and purity but is primarily determined by tumor-intrinsic gene expression. As recommended by the Reviewer, we have included the genes driving each cluster in Supplementary Table 4.

Figure R3

Fig 4. Too many IGV screen shots. Panels A,B,C suffice to make the main points. D could be in the supplement.

We agree, and have moved Fig. 4d to the supplementary data (Supplementary Fig. 4).

Fig 5 b/c. Remove background grid from the figures. Fig. 5 C/D legend is incorrect. Check text for correct call outs. Fig 5 E, What types of mutation are included in the % mutation figures (they don't match Fig 1C.) Inclusion of MAX: Really, MAX does not seem to be a major player, and it is not clear that it deserves emphasis at this low frequency (N of 1). Without more data to validate MAX the importance of this gene as a mutational target is uncertain.

We have modified Fig. 5b and 5c as suggested. We have corrected the legends to Fig. 5c and 5d; thank you for bringing this oversight to our attention.

The percentages given in Fig. 5e represent the entire spectrum of genetic alterations (single-nucleotide variants, small insertions/deletions, copy number alterations, fusions, microalterations, aberrant expression) underlying direct or indirect inactivation of *TP53*, *RB1*, and *PTEN* in LMS tumors. We have amended the legend to Fig. 5c to explain this more clearly.

We acknowledge that *MAX* mutations are not common in LMS, unlike, for example, in sporadic and syndromic gastrointestinal stromal tumors (Schaefer et al. Nat Commun 8:14674, 2017). Our intention, rather, was to demonstrate that in a subset of patients, inactivation of *RB1* is likely achieved through

other mechanisms than disruption of the *RB1* locus itself, namely loss of *CDKN2A* expression and subsequent overexpression of *CCND1* ($n = 3$) or acquisition of a pathogenic *MAX* mutation (Comino-Mendez et al. Nat Genet 43:663-667, 2011) and subsequent overexpression of *CDK4* and *CCND2* ($n = 1$; Fig. 5b of the revised manuscript). We hope the Reviewer will agree that presenting these alternative mechanisms, albeit rare, in the manuscript helps establish the point that loss of *RB1* function is a near-ubiquitous hallmark of LMS development.

ALT. Since the authors make a specific claim that this is the highest frequency of ALT observed, there needs to be more detail as to the justification of the cutoffs used. As the assays give a continuous output, there are always a number of samples falling near the cutoffs. Notably, normal blood and tumor arise from different lineages which need not have the same “normal” telomere repeat content. This can cause misinterpretation of the telomere content.

We completely agree with the Reviewer that it is difficult to use the telomere content of patient-matched blood samples as reference for the telomere content of tumor specimens, and that an ideal control would be normal tissue corresponding to a tumor’s cell of origin. However, as in most cancer genome sequencing studies, this material is not available and would be impossible to acquire, as most LMS tumors do not arise in a homogeneous anatomic region such as a parenchymatous organ, but deep in the retroperitoneum or the soft tissue of the extremities, where they are embedded in various “foreign” tissues.

Since we and others have observed a good correlation between the telomere content of ALT-positive tumor samples after normalization to matched blood samples, as exemplified by our data on pediatric glioblastoma (Deeg et al. bioRxiv 129106; doi: <https://doi.org/10.1101/129106>), we used the same strategy for the analysis of LMS tumors. It is indeed surprising that tumor telomere content does not correlate with the presence of ALT in this set of samples. To exclude that this might be attributed to specific features of the blood samples used as controls, we have revised Fig. 6b and included an additional plot in which telomere content was normalized to a single-copy gene. Notably, there is still no correlation between telomere content and ALT status. This is stated in the revised manuscript, and we now also point out that, in contrast to other tumor entities, telomere content does not appear to be a good ALT marker in LMS.

More generally, we would like to note that high telomere content is not necessarily a functional ALT hallmark since it can arise from very different mechanisms, most of which remain elusive. For example, initially very long telomeres are found in cells with the “ever-shorter-telomeres” phenotype in the absence of ALT (Dagg et al. Cell Rep 19:25442556, 2017). Furthermore, telomere length is the result of the interplay between various processes; for example, the capacity of an ALT-positive tumor to maintain telomere length may be exceeded by a very high proliferation rate, resulting in net telomere shortening. To precisely determine the frequency of ALT in LMS tumors, we have employed the C-circle assay, which provides a direct quantitative measure of this process through detection of telomere-derived, partially double-stranded circles of DNA that are currently regarded as the most specific ALT marker (Henson et

al. *Nat Biotechnol* 27:1181-1185, 2009; Cesare and Reddel. *Nat Rev Genet* 11:319-330, 2010; Oganessian and Karlseder. *Mol Cell* 42:224-236, 2011).

The C-circle while reliable when strong is not necessarily easy to evaluate at low signals. Were there repeat assays for each tumor or only 1?

Due to the limited amounts of DNA from primary human LMS samples, the C-circle assays shown are from one experiment. However, we would like to note that we have carefully calibrated our assay conditions in titration experiments with cell lines, and that several cell line controls were included in the experiments with LMS tumors to determine the signal ratio used for classifying samples as ALT-positive. To illustrate this more clearly, we have revised Fig. 6a to include U2OS and HeLa cells as references, which were measured several times together with the LMS samples. It is certainly conceivable that the samples with weak C-circle signals include false negatives, which would result in underestimation of the frequency of ALT. In contrast, we consider it highly unlikely that our analysis includes any false positives, and are therefore confident that our conclusion regarding the high frequency of ALT in LMS is well-founded.

Also, while the approach taken to investigate telomere associated genes is interesting in an abstract sense, it is very hard to interpret Fig 6 C since these genes have diverse roles with varying degrees of mechanistic connection to telomere biology. In general, as CNAs are the principal drivers of this figure, this problem devolves to the challenging interpretation of the CNA plots in Fig 2A. Since a large portion of the genome has frequent CNAs, then any large set of genes (many hundreds are listed in TELNET) will generate positive results. This is a great way to generate hypotheses, but doesn't really prove much. Also, since *ATRX* is on X, copy number is gender biased, *ATRX* CNA/LOH may not always be meaningful. In addition, I am not sure that amplification deserves to be added into the relevant mutation list for this (or any other gene) whose loss is associated with ALT.

As pointed out by the Reviewer, our work has allowed us to nominate novel candidate drivers of ALT whose precise roles in telomere maintenance need to be investigated further. We consider these findings an important step towards understanding the molecular mechanisms underlying ALT in human cancer. In current pan-cancer analyses by The Cancer Genome Atlas Research Network (Barthel et al. *Nat Genet* 49:349-357, 2017) and the International Cancer Genome Consortium (Sieverling et al. *bioRxiv* 157560; doi: <https://doi.org/10.1101/157560>), ALT-positive cases are identified based on mutations in *ATRX* or *DAXX* and high telomere repeat content. We now demonstrate that ALT occurs in a large fraction of LMS cases with wild-type *ATRX/DAXX* and low telomere content, pointing to the existence of telomere maintenance mechanisms that differ from "classical" ALT in terms of potential driver genes and telomere repeat content.

Based on our observations, which may have considerable implications for future cancer genomic studies as they challenge the prevailing view of how ALT-positive cases can be identified, we have begun to explore the role of non-canonical activation of ALT in the pathogenesis of LMS, which potentially involves

disruption of genes such as *RBL2* and *SP100*. These experiments are challenging due to the lack of ALT-positive LMS cell lines. Furthermore, the induction ALT, a complex multi-step process, by RNA interference-mediated gene knockdown or CRISPR/Cas9-mediated gene knockout has not been successful for us and several other groups, as established cancer cell lines are either captured in replicative senescence or upregulate telomerase expression before they activate ALT. Our current efforts in the laboratory therefore include the establishment of LMS cell lines and xenografts from fresh patient material, which will hopefully also yield ALT-positive models that can be used, in the intermediate to long term, to study the mechanisms underlying the exceptionally high frequency of ALT in LMS tumors. We hope the Reviewer will agree that in-depth mechanistic studies on the molecular underpinnings of *ATRX/DAXX*-independent ALT are beyond the scope of the current report, which aimed to provide the first high-resolution genomic and transcriptomic map of LMS.

We agree with the Reviewer that hemizygous loss of *ATRX* may have different functional consequences in men vs. women, even though there is evidence to support that this locus is dosage-sensitive (ClinGen Haploinsufficiency Score of 3; <https://www.ncbi.nlm.nih.gov/projects/dbvar/clingen>). In any case, we would like to re-emphasize that ALT occurs in a large proportion of LMS tumors with wild-type *ATRX*, and that this mechanism should therefore not be evaluated in the context of *ATRX* alterations alone.

The purpose of Fig. 6c was to provide an overview of the entire spectrum of genetic alterations affecting telomere maintenance genes in LMS tumors. We completely agree with the Reviewer that for candidate genes whose inactivation leads to ALT, the focus should be on deleterious alterations, as in the case of our top candidates *RBL2* and *SP100*, which are almost exclusively affected by copy number losses. However, in the case of other telomere maintenance genes, e.g. *BLM* and *ERCC4*, higher expression is associated with ALT. To address the Reviewer's point, we have explained this more clearly in the manuscript text, but would suggest to leave Fig. 6c unchanged for the sake of completeness.

“BRCAness” Fig 7 A. Essentially the same problem applies to this plot as to Fig 6 C. This is a fine way to explore the data, but is insufficient to establish the point. Regarding 7B, please provide the selected YASPA cut-off values, as the analysis could not be reproduced with the provided data. Furthermore, please provide significance values for the analysis. It is not clear if the pattern is random or statistically significant. In general, signature it does not seem to be the dominant pattern in most instances. The stacked barplots do not help. While it is absolutely true that signature 3 is associated in breast/ovarian cancers with impaired homologous recombination, the absolute strength of the signature in most other cancers is much weaker than in those cancer and may be indicative of other pathways with similar end effects. Similarly, while I am sure that the data showing some degree of sensitivity of cells to olaparib is correct, this does not always correlate with meaningful clinical sensitivity. While appealing to connect LMS instability to a drug target, it could be misleading to propose defective HR as a target since it is not firmly established by the available data. Delving into the many aspects of DNA damage and repair which could be disturbed in LMS would require much more analysis supported by mechanistic experimentation in model systems.

We thank the Reviewer for these helpful suggestions.

To ensure that readers can reproduce our signature analyses, the YAPSA cut-off values are now provided in the Methods section.

To address the Reviewer's question regarding statistical significance, we have taken two steps. First, we now provide confidence intervals based on likelihood ratio tests. Second, we have compared the mutational signatures extracted from the LMS cohort to those extracted from a background of 7,042 cancer samples (whole-genome sequencing, $n = 507$; whole-exome sequencing, $n = 6,535$; corresponding to the dataset that formed the basis for the original discovery of mutational signatures [Alexandrov et al. Nature 500:415-421, 2013] and was used by us to train the YAPSA cut-off values) using Fisher exact tests corrected for multiple comparisons according to the Benjamini-Hochberg method. The resulting P values, which have been included in the revised manuscript, demonstrate the significant enrichment of signature Alexandrov-COSMIC 3 associated with defective homologous recombination repair of DNA double-strand breaks in LMS tumors ($P = 2.67 \times 10^{-30}$).

We agree with the Reviewer that the graphical representation of our signature analyses was not optimal. To better visualize the contribution of different signatures to the mutational catalogs of individual LMS tumors, we have split the stacked barplots into separate tracks (one per signature) with single barplots, in which the confidence intervals described above can be displayed.

More generally, we would like to stress that we have subjected a cohort of primary human LMS samples as well as LMS cell lines to a comprehensive analysis of established genomic imprints of homologous recombination deficiency, including specific patterns of structural rearrangements, alterations of individual genes known to be involved in homologous recombination based on detailed mechanistic experimentation in different tumor entities, and enrichment of specific mutational signatures that have been proposed as surrogate marker of defective homology directed double-strand DNA break repair (Alexandrov et al. Nat Commun 6:8683, 2015; Lord and Ashworth. Nat Rev Cancer 16:110-120, 2016; Davies et al. Nat Med 23:517-525, 2017). This approach goes far beyond the genetic criteria that are usually applied in patients with epithelial cancers to determine their sensitivity to PARP inhibitor treatment (see Mateo et al. N Engl J Med 373:1697-1708, 2015 for a recent example). Furthermore, we have shown that olaparib and cisplatin treatment impair the growth of LMS cell lines whose genomic profiles resemble those of primary LMS patient samples, including damaging alterations of multiple genes that have been associated with defective homologous recombination and induction of synthetic lethality to PARP inhibition in the recent literature. In our view, these data represent compelling evidence that the concept of "BRCAness" as potentially actionable genomic feature extends to LMS.

In addition, we wish to emphasize that the concept of mutational signatures is universal as it stems from an extensive cross-entity analysis and the signatures are linked to mutational mechanisms independent of tissue context. To our knowledge, there also is no evidence to suggest that the relative contribution of a specific signature to a tumor's mutational catalog is a critical determinant of an ensuing synthetic lethal relationship, which should be linked to the presence of an underlying mutational process, such as defective homologous recombination, irrespective of concomitant processes that may be reflected in

additional signatures. These considerations notwithstanding, we are fully aware that molecular mechanism-guided clinical trials will be necessary to determine whether the genomic features of LMS translate into meaningful sensitivity to pharmacologic PARP inhibition, either alone or in combination with DNA-damaging cytotoxic agents. In this regard, we would like to note that our findings have spurred the development of a multi-institutional clinical trial evaluating the efficacy of olaparib in combination with trabectedin in patients with advanced-stage cancer – including LMS, an aggressive malignancy for which no targeted therapy exists – and defective homologous recombination as assessed using a biomarker that integrates information on genomic instability, functional mutations in homologous recombination pathway genes, and mutational signatures (ClinicalTrials.gov Identifier NCT03127215).

As alluded to by the Reviewer, we have not mechanistically evaluated the homologous recombination pathway, e.g. by directly assessing RAD51 foci formation via immunofluorescence or other DNA repair complexes via immunohistochemistry. Such experiments would be technically complex and difficult to standardize as they would require viable LMS samples to be exposed to DNA damage *ex vivo* before functional biomarkers of defective homologous recombination can be evaluated, and we hope the Reviewer will agree that this task is beyond the remit of the current study, which aimed to determine the genomic and transcriptomic landscape of LMS.

REVIEWERS' COMMENTS:

Reviewer #1 (Remarks to the Author):

I have reviewed the rebuttal with the revised manuscript. I believe all points have been adequately addressed in great detail and accept the limitations described for data supporting "BRCAness"

Reviewer #2 (Remarks to the Author):

The authors have responded adequately with this revision to most of the points raised in the previous review.

The issue of PARP sensitivity is still weakest aspect of this paper (presenting only photographs of drug treated plates without IC50 determination does not match usual standards and also does not fall in the nM sensitivity typically seen in BRCA2 deficient breast cancer cell lines). There is still considerable room for future research to understand the pharmacogenomics of LMS with respect to PARP inhibitors. Despite the authors' strong argument in favor of the relevance the concept of "BRCAness", the fact remains that this concept grew out of research in BRCA1/2 deficient epithelial cancers and even in that system is not linked in mechanistic detail to the development of AC-3. Biallelic BRCA2 deficiency is associated with AC-3, but the converse is not proven. AC-3 is actually relatively common, and may well arise in genomes which are highly rearranged as a result of a variety of defects which favor NHEJ and related processes over HR yet may not result in uniform PARP inhibitor sensitivity. I do not argue that the authors' proposal is incorrect, simply that it lacks sufficient evidence. I recognize that others are also using the concept of "BRCAness" loosely but I do not feel that the link between the mutational profile and the minimal in vitro experimentation is sufficient to definitively put this label on LMS. I would be much more comfortable with softened language on this point and a clear acknowledgment that additional biochemical studies would be necessary to definitely prove the claim of HR deficiency.

RE: NCOMMS-17-15211A
Integrative genomic and transcriptomic analysis of leiomyosarcoma

We thank the Reviewers for their insightful and constructive comments, which have further improved this work. Please find enclosed a revised manuscript that has been modified in accordance with their recommendations. Our specific responses to the Reviewers' comments are detailed individually below.

Reviewer #1

I have reviewed the rebuttal with the revised manuscript. I believe all points have been adequately addressed in great detail and accept the limitations described for data supporting "BRCAness".

We are grateful for these favorable comments.

Reviewer #2

The authors have responded adequately with this revision to most of the points raised in the previous review.

The issue of PARP sensitivity is still weakest aspect of this paper (presenting only photographs of drug treated plates without IC50 determination does not match usual standards and also does not fall in the nM sensitivity typically seen in BRCA2 deficient breast cancer cell lines). There is still considerable room for future research to understand the pharmacogenomics of LMS with respect to PARP inhibitors. Despite the authors' strong argument in favor of the relevance the concept of "BRCAness", the fact remains that this concept grew out of research in BRCA1/2 deficient epithelial cancers and even in that system is not linked in mechanistic detail to the development of AC-3. Biallelic BRCA2 deficiency is associated with AC-3, but the converse is not proven. AC-3 is actually relatively common, and may well arise in genomes which are highly rearranged as a result of a variety of defects which favor NHEJ and related processes over HR yet may not result in uniform PARP inhibitor sensitivity.

I do not argue that the authors' proposal is incorrect, simply that it lacks sufficient evidence. I recognize that others are also using the concept of "BRCAness" loosely but I do not feel that the link between the mutational profile and the minimal in vitro experimentation is sufficient to definitively put this label on LMS. I would be much more comfortable with softened language on this point and a clear acknowledgment that additional biochemical studies would be necessary to definitely prove the claim of HR deficiency.

We agree with these comments, and have amended the manuscript to acknowledge more clearly that additional preclinical experimentation as well as genomics-guided clinical trials will be necessary to definitely establish whether the molecular characteristics of LMS – which are not limited to signature AC3, but also include frequent alterations in homologous recombination DNA repair genes and multiple

structural rearrangements – are reflective of homologous recombination deficiency and translate into sensitivity to PARP inhibitor treatment in vivo. Specifically, we have expanded the fourth paragraph of the Discussion section and toned down the language on the issue of “BRCAness” as potential liability of LMS tumors throughout the manuscript (Introduction, last paragraph; Results, last paragraph; Discussion, last paragraph).